# Contrasting seasonal patterns in particle aggregation and DOM transformation in a sub-Arctic fjord

Maria G. Digernes*[1], Yasemin V. Bodur*[2], Martí Amargant-Arumí[2], Oliver Müller[3], Jeffrey A. Hawkes[4], Stephen G. Kohler[1], Ulrike Dietrich[2], Marit Reigstad[2], Maria Lund Paulsen[5]

[1]Department of Chemistry, Norwegian University of Technology, Trondheim, 7049, Norway
[2]Arctic and Marine Biology, UiT The Arctic University of Norway, Tromsø, 9019, Norway
[3]Department of Biological Sciences, University of Bergen, Bergen, 78303, Norway
[4]Department of Chemistry, University of Uppsala, Uppsala, 75124, Sweden
[5]Department of Biology, Aarhus University, Aarhus, 8000, Denmark

*Correspondence to*: maria.g.digernes@ntnu.no, yasemin.bodur@uit.no. *these authors have contributed equally to this work.

**Abstract.** Particulate (POM) and dissolved (DOM) organic matter in the ocean are important components of the Earth's biogeochemical cycle. The two are in constant dynamic change through physical and biochemical processes, however, they are mostly treated as two distinct entities, separated operationally by a filter. We studied the seasonal transition of DOM and POM pools and their drivers in a sub-Arctic fjord by monthly environmental sampling and performing experiments at selected time-points. For the experiments, surface water (5 m) was either pre-filtered through a GF/F filter (0.7 µm), or left unfiltered, followed by 36 h incubations. Before and after the incubation, samples were collected for dissolved and particulate organic carbon concentrations (DOC, POC), extracellular polymeric substances (EPS), microbial community (flow cytometry), and the molecular composition of DOM (HPLC-HRMS). During the biologically productive period, when environmental POC concentrations were high (April, June, September), the filtered water showed an increase of POC concentrations . While POC concentrations increased in September, DOM lability decreased based on changes in average hydrogen saturation and aromaticity of DOM molecules.  In contrast, during the winter period (December, February), when environmental POC concentrations were low, lower concentrations of POC were measured at the end of the experiments compared to the start. The change in POC concentrations was significantly different between the biologically productive period and the winter period (t-test; $p < 0.05$). Simultaneously, the DOM pool became more labile during the incubation period, as indicated by changes in average hydrogen saturation, aromaticity, and oxygen saturation with implications for carbon cycling. The change in POC was not directly associated to an antagonistic change in DOC concentrations, highlighting the complexity of organic matter transformations, making the dynamics between POC and DOC difficult to quantify. However, in both periods, bacterial activity and EPS concentrations increased throughout the incubations, showing that bacterial degradation and physical DOM aggregation drive the transformations of POM and DOM in concert, but at varying degrees under different environmental conditions.

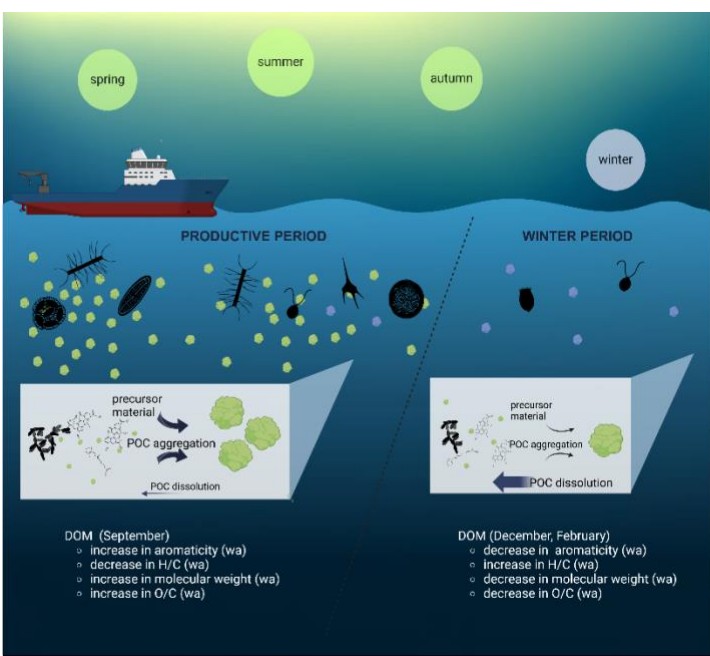

## 1 Introduction

Dissolved (DOC) and particulate organic carbon (POC) play an important role in earth's biogeochemical cycling, and their availability and variation are strongly driven by seasonal ecosystem dynamics. In the ocean, dissolved organic matter (DOM) makes up 97% of total organic matter, while only 3% is in particulate form (Hansell et al., 2009). Marine DOM is one of the largest stocks of organic carbon on Earth, contributing to long term carbon storage in the ocean's interior. Particulate organic matter (POM) also aids in carbon sequestration by sinking to the seafloor and transporting carbon from the ocean surface (Iversen, 2023; Turner, 2015). POM (usually defined as $> 0.4 – 0.7$ µm until several mm in size) is mostly composed of protist cells, fecal pellets, biogenic leftovers such as mucilaginous feeding nets and DOM that stick together to form large marine snow aggregates. DOM and POM are separated operationally merely by size (nominal filter pore size between 0.2 and 0.7 µm; Carlson et al., 2015) and often studied independently, despite the constant dynamic change between the two fractions (He et al., 2016; Verdugo et al., 2004; Wells, 1998). DOM does not sink gravitationally; however, it can contribute to the export of biogenic carbon by downward mixing (Hopkinson & Vallino, 2005), aggregation to POM, or formation of a sticky matrix for particles (Engel et al., 2004; Hansell et al., 2009; Iversen, 2023; Wells, 1998). Consequently, DOM provides a particle source which is well known, but often overlooked in ecosystem studies (Engel et al., 2004).

DOM is primarily generated and secreted by phytoplankton during their growth as they release 2 – 50% of the photosynthetically fixed carbon as DOM (Thornton, 2014; Paulsen et al., 2018). DOM can also be produced by zooplankton

during grazing and excretion, as well as by bacterial and viral processes such as lysis and excretory release and through the dissolution of particles (Carlson & Hansell, 2015; Riley, 1963; Wagner et al., 2020). Freshly produced labile DOM exhibits temporal fluctuations in accordance with the seasonal and spatial variations in phytoplankton abundance, microbial communities, and inorganic nutrient availability (Osterholz et al., 2014; Paulsen et al., 2019; Retelletti Brogi et al., 2019; Vernet et al., 1998). Labile DOM, which constitutes less than 1% of the overall DOM reservoir, displays relatively short turnover times, typically ranging from hours to days (Hansell, 2013). Conversely, semi-labile and recalcitrant DOM persists in the ocean over more extended time scales, ranging from months to millennia (Fleurs et al., 2012; Hertkorn et al., 2006). Efforts have been made to characterize the seasonal DOM pool; however, the bulk of DOM characterization studies are mostly conducted in spring and autumn (Flerus et al., 2012; Osterholz et al., 2014; Retelletti Brogi et al., 2019) and thus present a need for winter DOM characterization studies.

Depending on physicochemical and biological conditions, DOM can undergo various transformations, such as aggregation, dissolution, adsorption, resorption, autoxidation, (photo)chemical and biological degradation (Carlson & Hansell, 2015). These transformations result in alterations of DOM molecular composition, abundance, and size, with implications for the ecosystem and the carbon cycle. DOM that is channelled through the microbial loop can be transformed to more recalcitrant forms (Jiao et al., 2010) or converted back to carbon dioxide, while particles that are formed through DOM aggregation can potentially sink and lead to carbon export. Dissolution of POM to DOM removes OM from the classical food web, decreases the particulate pool, leads to a longer turn-over time of OM in the water column and decreases the sinking potential. Despite the strong dependency of DOM and POM transformations on ecosystem processes, not much is known about changes in the DOM-POM continuum under contrasting environmental and seasonal conditions.

Precursor material such as colloids or gels act as the "gray zone" between the dissolved and the particulate fractions (Orellana & Leck, 2015). They can be characterized as high molecular weight (HMW) DOM (> 1 kDa) but are also often quantified as particulate material as they can remain on filters due to their sticky, flexible properties. More than ¼ of oceanic DOC can be in colloidal form (Kepkay, 1994). Transparent exopolymer particles (TEP) are sticky gels formed from extracelluar polymeric substances (EPS) composed of carbohydrate-rich phytoplankton exudates (Passow et al., 1994; Passow, 2002b). Spontaneous assembly of smaller molecules can lead to the abiotic formation of gels (Chin et al., 1998; Passow, 2000). These processes can be triggered by small changes in ambient pH, ionic concentration, temperature, or light (He et al., 2016; Timko et al., 2015; Verdugo et al., 2004). Colloids/gels can aggregate to particulate matter through chemical coagulation or physical flocculation (Engel et al., 2004). These aggregated particles can form again after filtration of sea water indicated by the presence of POC in dissolved samples at filtration timescales (Riley, 1963; Sheldon et al., 1967; Valdes Villaverde et al., 2020; Xu & Guo, 2018) and their measurement often appear as an analytical artifact.

High latitude ecosystems are characterized by a strong seasonality, creating contrasting environmental conditions (Petersen & Curtis, 1980). Arctic and sub-Arctic fjords can be periodic hotspots for biological productivity, and locations of high carbon production, turnover and export. With the beginning of spring, the return of sunlight and replenished nutrients in the upper water column fuel intense phytoplankton blooms, and with that the production of fresh POM and DOM (Paulsen et al., 2018; Walker et al., 2022; Wetz & Wheeler, 2007). Spring is influenced by autochthonous labile DOM production composed of low oxygen and high hydrogen saturation (Hansell, 2013). Late spring and early summer are subject to high freshet with allochthonous DOM input from river sources composed of oxygen rich DOM compounds (Koch et al., 2005; Sleighter & Hatcher, 2008). Towards summer, nutrients become depleted at the surface, and the system is dominated by heterotrophic processes and increasing carbon turnover (Carlson et al., 2015; Repeta, 2015). With decreasing nutrient concentrations and increasing abundance of senescent cells, EPS can be excreted at high concentrations and trigger flocculation events during post-bloom conditions in summer (Alldredge & Gotschalk, 1989; Engel, 2000; Hellebust, 1965; Mague et al., 1980; Mari & Burd, 1998; Myklestad, 1995; Passow, 2002a; Thornton, 2014). In Autumn, DOM is converted to more recalcitrant compounds with lower hydrogen saturation (Osterholz et al., 2014) and there is a decrease in TEP (von Jackowski et al 2020). Autumn mixing in fjords can redistribute nutrients in the water column and fuel autumn blooms later in the year (Vonnahme et al., 2022). During winter, low light conditions limit primary production and the water column is subjected to an increase in vertical mixing. Organic matter concentrations are at their lowest during this period and seemingly recalcitrant due to dissolved organic carbon accumulation (Hansell, 2013). However, microbial degradation of organic matter can still take place (Vonnahme et al., 2022; Wietz et al., 2021).

Studies of the DOM–POM continuum in aquatic environments have mostly been interpreted either from an ecological or a chemical point of view. We are aware of some studies with laboratory and field observations in rivers (Attermeyer et al., 2018; Keskitalo et al., 2022), permafrost thaw (Shakil et al., 2021), Atlantic coast (Riley, 1963) and Pacific coast (Sheldon et al., 1967), or focused on the associated changes in the DOM–POM continuum in boreal peatlands (Einarsdóttir et al., 2020) and rivers (Xu & Guo, 2018). The overarching aim of this study is to document the effect of DOM–POM processes under contrasting ecological conditions and shed light on the involved processes. This is relevant in a highly seasonal and drastically changing Arctic, to understand possible implications for the carbon cycle and carbon burial. We hypothesize that the biologically active period with higher POM and DOM concentrations has a higher potential for aggregation of DOM in comparison to the winter period. To test the hypothesis, we closely followed the DOM and POM concentration and characteristics along with a range of environmental parameters during a full annual cycle in the surface water of a sub-Arctic fjord. Additionally, we investigated the behaviour of extracellular polymeric substances and its link to biological processes and DOM aggregation. We designed experiments to examine the partitioning between DOM and POM, by incubating 1) filtered (0.7 μm) fjord water, where biological activity was strongly reduced and particles removed, and 2) unfiltered water where pre-formed gels, microorganisms and other POM sources were kept as under natural conditions. This allowed us to simultaneously obtain an insight into the net-aggregation processes in the filtered water and the sum of the

aggregation,dissolution and biological processes in the unfiltered controls during contrastingseasons. We further aimed to understand a) how the bioavailability of the vast DOM pool was affected by seasonal transformations of the POM pool , and b) how biological processes impacted the DOM-POM continuum.

## 2 Methods

### 2.1 Sampling of standard and experimental parameters

Fieldwork was conducted monthly in the mouth of an Arctic fjord close to Tromsø/Romsa (Ramfjorden/Gáranasvuotna; 69°31'34"N, 19°1'33.0"E) between 16.09.2020 and 24.08.2021 with RV Hyas (Fig. 1). Ramfjorden is a sidearm of the larger Balsfjorden, which is mainly influenced by Norwegian Coastal Water, and seasonally by the inflow of Atlantic Water in spring (Eilertsen et al., 1981). Ramfjorden receives allochthonous input mainly through the inflow from two rivers (Storelva and Sørbotnelva). During winter, the inner part of the fjord freezes and is covered by sea ice (O'Sadnick et al., 2020). Sampling for this study occurred in the mouth of the fjord, which was ice-free year-round. In the beginning of each sampling event, a CTD equipped with fluorescence, oxygen and turbidity sensors was deployed. Water samples were taken at 5m depth with a GoFlo (General Oceanics, 20 L). Due to lack of research vessel access in July, we sampled in the beginning and the end of June (08. and 30.06.2021) and hence refer to these sampling events throughout the manuscript as "June" and "July".

From the GoFlo, 15 L were subsampled into a plastic canister for later processing in the lab every month (Fig. 2a-c) for the following biogeochemical parameters: chlorophyll-a (Chl-a), total particulate matter and particulate inorganic matter (TPM and PIM), protist taxonomy, extracellular polymeric substances (EPS) and flow cytometry (FCM). For dissolved organic matter characterization (DOM), water from the same GoFlo was subsampled into a separate acid-washed 3 L canister. Three triplicate samples for dissolved organic carbon (DOC) and total dissolved nitrogen (TDN) were taken with muffled glass vials and for nutrients with 50 mL falcon tubes directly from the GoFlo. All tubing used was acid-cleaned prior to use, and sample containers were pre-rinsed with sample water before sampling.

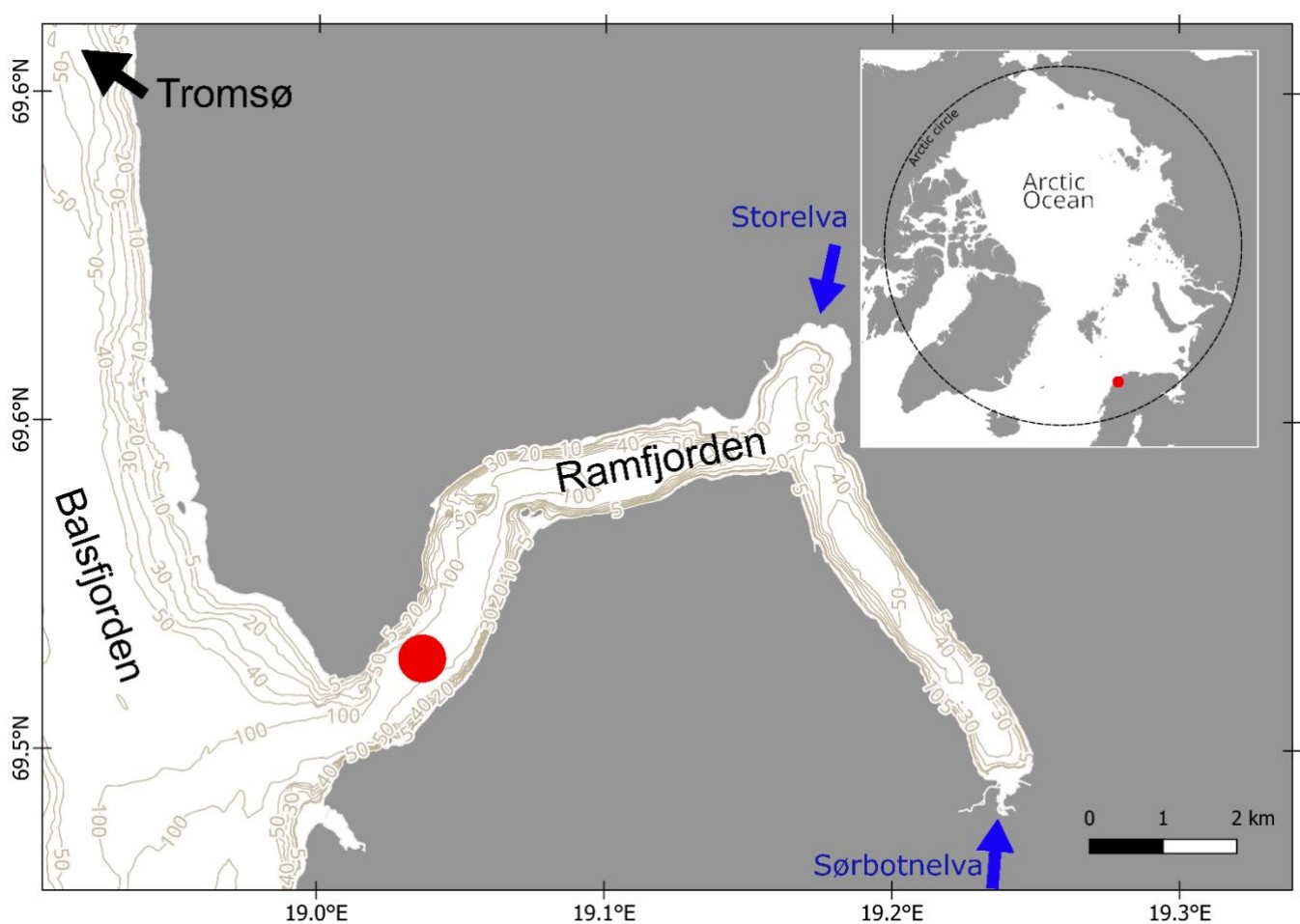

**Figure 1: Position of Ramfjorden/Gáranasvuotna within the Arctic Circle and sampling location (red dot) within the fjord.** Land area and depth contours of Ramfjorden were retrieved from kartkatalog.geonorge.no.

## 2.2 Aggregation experiment

To demonstrate seasonal contrasts of the aggregation potential of DOM in the fjord, experiments were conducted every two months (Fig. 2d-f). The water from three GoFlos was filtered through a 90 µm mesh to remove large grazers and evenly distributed among 3 acid-washed 20 L canisters after being pre-rinsed with sample water through staggered filling. The canisters were covered with black plastic bags to minimize light exposure. After bringing them on shore, the canisters were stored in a controlled temperature room kept dark at 5°C where the experiments were carried out by using headlamps with red light to reduce the possibility of biological production. Each time prior to the experiments, all surfaces and the floor in the cold room were washed with Citranox® acid detergent and distilled water to remove dust and minimize carbon contamination. Any handling of the samples was carried out wearing microporous laminated clean suits (Tyvek®, DuPont, USA), and all used equipment and tubing was acid-rinsed (plastic equipment), combusted (glass equipment) or cleaned in an ultrasonic cleaner (metal equipment) prior to use.

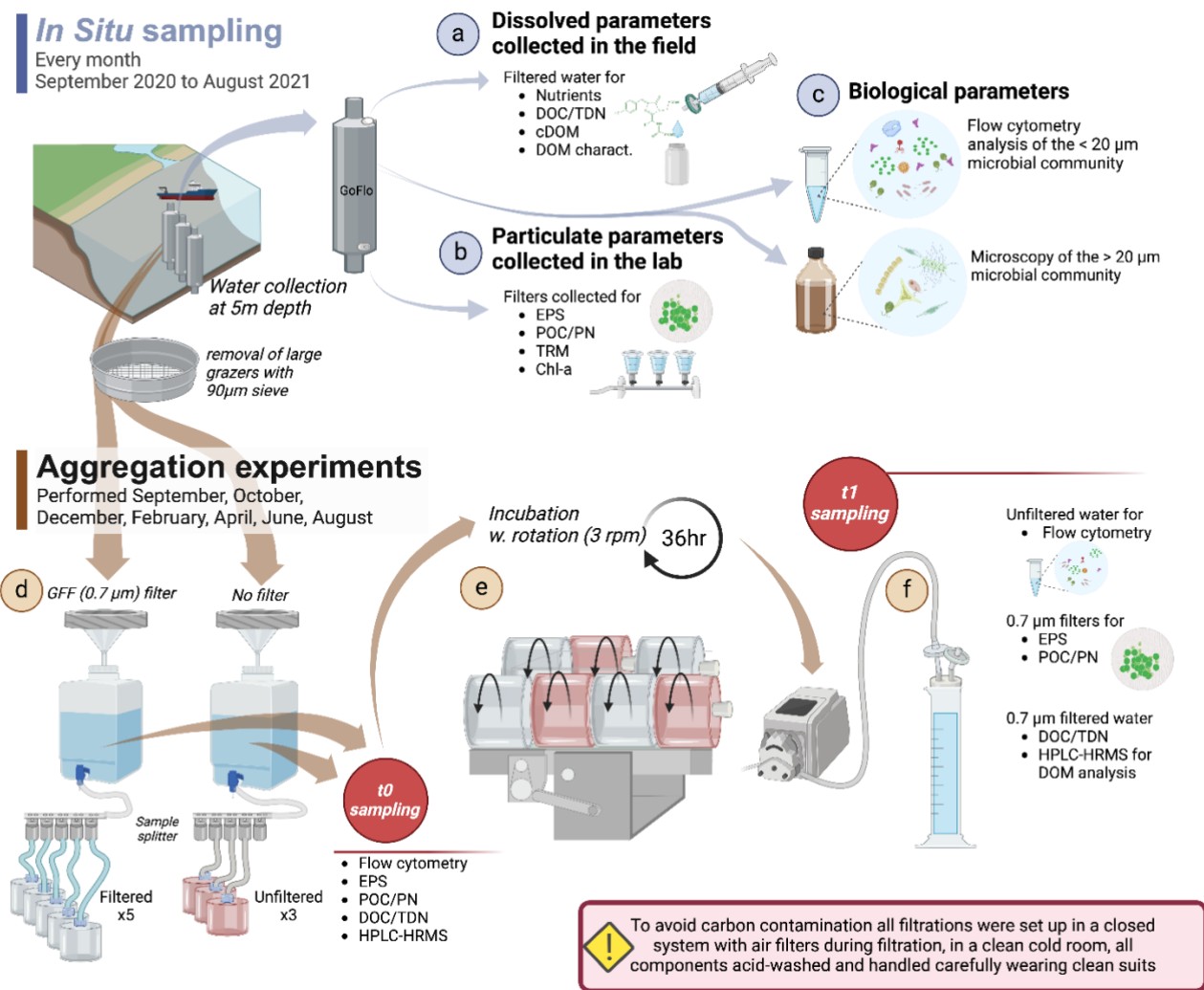

**Figure 2: Sampling scheme and experimental setup.** Water was sampled with 20L GoFlos at 5 m depth, and the following standard parameters (STD) were collected: (a) nutrients, dissolved organic carbon (DOC), total dissolved nitrogen (TDN), dissolved organic matter characterization (DOM charact.), and colored DOM (cDOM) were taken directly in the field; (b) filters for the analysis of extrapolymeric substances (EPS), particulate organic carbon and particulate nitrogen (POC/PN), total particulate matter (TPM) and Chlorophyll-a (Chl-a) were collected in the lab; and (c) water samples were preserved for protist taxonomy and flow cytometry (FCM). The experimental water was collected from 3 x 20L GoFlos, funnelled through a 90µm sieve and distributed equally among 3 canisters. The experiment was set up as in (d): The experimental water was channelled through a pressure-filtration system with a GF/F filter (filtered; F) and without a GF/F filter (unfiltered; UF), and then sampled respectively for EPS, FCM, DOM, DOC, TDN and POC were taken. Afterwards, the F and UF water respectively were distributed evenly among roller tanks with a sample splitter. (e) During a 36h incubation period, the tanks were rolled at 3rpm. (f) After the termination of the incubation, each tank was subsampled for EPS and FCM. Subsequently, the tanks were connected to a peristaltic filtration system with in-line filters that were used for the sampling of POC. The filtrate was collected in graduated cylinders, which was then subsampled for DOC and DOM characterization.

The water from two canisters was pressure-filtered through a single layer (September, October, December) or a double layer (February, April, June) of pre-combusted GF/F filters (0.7 µm, Whatman, diameter: 130 mm) which was placed on an acid-washed plexiglass filter holder and collected in another canister (filtered, F; Fig. 2d). The effect of a double or a single layer on POC concentrations was tested in January, for the results and a short discussion see the supplementary (Fig. S1). To compare the aggregation behavior of the filtrate to the behavior of the seasonally changing natural unfiltered water, we left the water from the third canister unfiltered (UF). The water from the third canister was streamlined through the filtration system in the same way, but without a filter, to account for possible hydraulic stress during the filtration procedure and treat the UF samples in the same way as F. Filtered treatment (F) samples were collected for POC/PN, EPS, FCM, DOC and DOM characterization immediately after this step (t0 samples). UF treatment samples for t0 were collected for POC/PN, EPS and FCM. It should be noted that for DOC and DOM characterization, UF treatment at t0 is equivalent to F treatment at t0. Also, in September the triplicate field samples of DOC were used as t0. Subsequently, the F and UF water was distributed evenly into cylindrical plexiglass tanks (1.8 L self-manufactured,18 cm inner diameter, 6 cm height) by using a splitter. The tanks were filled until the top and air bubbles were removed before they were closed with silicone stoppers. Eight tanks (3x UF, 5x F) were placed on two rolling tables in randomized order and rolled for 36 h at 3 rpm to ensure a homogenized distribution of the water body throughout the incubation (Fig. 2e).

After 36 h, the incubation was stopped, and samples (t1) were taken from each tank for EPS and FCM after gently homogenizing the tanks by turning them slowly 20 times. Afterwards, all tanks were connected air-tight (to minimize contamination) to a peristaltic filtration system where GF/F (Whatman®, 0.7 µm, pre combusted) filters were connected in-line and the filtrate was collected in acid-washed and equally air-tight graduated cylinders (Fig. 2f). After rinsing the whole system and the graduated cylinders with sample water, the filters were replaced and the subsequently collected filtrate was used for the subsampling of DOC and DOM. After each tank was emptied completely, the filters were collected for POC/PN analyses in the following way: the tubing of the lower part of the filter holder was removed, and any excess water in the tubing or in the filter holder was sucked onto the filter with a syringe. Subsequently, the dry filters were folded, packed into pre-combusted aluminum foil, and frozen at −20°C. The exact volume used for the subsampling of all the parameters was read from the graduated cylinders. EPS, FCM, DOC, DOM and POC/PN were analyzed as described in the following section.

After the experiment was finalized, all used equipment was soaked in acid for several hours, rinsed three times with MilliQ, dried in a drying oven at 60°C and finally stored in airtight zip bags or boxes until the next sampling to prevent any carbon contamination.

**2.3 Processing of samples**

In situ samples for particulate organic carbon (POC) and particulate nitrogen (PN) were filtered in triplicates, and experimental samples as described in Section 2.2, onto pre-combusted GF/F filters (0.7 µm, Whatman), packed in combusted aluminum foil

and frozen at −20°C. POC samples were dried for 24 h at 60°C, subsequently acid-fumed (HCl) in a desiccator for 24 h to remove all inorganic carbon, and finally dried again for 24 h at 60°C. The filters were transferred into tin capsules and measured with a CE440 CHN elemental analyzer (Exeter Analytical) and Acetanilide was used as standard.

Samples for the colorimetric determination of extracellular polymeric substances (EPS) were taken in situ between November 2018-September 2019, and then again between February-August 2021 in situ and for the experiment. We measured EPS instead of TEP directly, because the EPS measurement has a higher detection accuracy for all carbohydrates in a sample (including TEP + TEP precursors) compared to the Alcian Blue method after Passow & Alldredge (1995; Bittar et al., 2018; Li et al., 2018). The sampled water (150ml) was filtered onto 0.4 µm polycarbonate filters. Following the colorimetric method by Dubois et al. (1956), a mixture of phenol and concentrated sulfuric acid was used to extract material from the filter to determine total carbohydrates in the sample. A spectrophotometer (UV-6300PC, VWR) was used to measure the absorbance of the solution at 485 nm. Since concentrations were too low to be calculated reliably with a standard curve against xanthan gum concentrations, relative EPS concentrations are depicted as "absorption at 485 nm".

Flow cytometry was used for the determination of bacteria, virus, pico- and nano-sized phytoplankton abundances in situ and in the experiment. Unfiltered samples of 5 mL were fixed in duplicates with glutaraldehyde (0.5% final concentration) and frozen at −80°C until analysis within 3 months. The samples were thawed and pico- and nanophytoplankton were analyzed directly on an Attune® Acoustic Focusing Flow Cytometer (Applied Biosystems by Life Technologies). The populations of phytoplankton were grouped based on their pigmentation on biplots of green vs. red fluorescence. Before counting bacteria and viruses, the DNA was stained with SYBR-green I and groups were discriminated on biplots of side scatter vs. green fluorescence. Actively dividing bacterial cells contain more DNA, therefore the ratio of High Nucleic Acid (HNA) bacteria to Low Nucleic Acid (LNA) bacteria is here used as an indicator of the relative activity of the bacterial community. The following conversion factors were used to convert various microorganism groups to carbon (pg C cell$^{-1}$): bacteria (0.02), *Synechococcus* sp. (0.29), pico-eukaryotes (0.57), nanophytoplankton (7).

Samples for dissolved organic carbon (DOC) and total dissolved nitrogen (TDN) were filtered on GF/F filters (0.7 µm, Whatman, pre combusted) and acidified to pH 2 (HCl, double distilled, AnalaR® NORMAPUR®, VWR chemicals). Samples were stored at 6°C until analysis. DOC and TDN determination were executed via high temperature catalytic oxidation method using a Total Organic Carbon Analyzer (TOC-L CPH/CPN™, Shimadzu). Potassium hydrogen Phthalate (KHP, Merck) was used for external calibration. Seawater reference samples from the University of Miami (Hansell research laboratory) were analyzed throughout sample runs (repeatability > 95%, n = 78).

In situ nutrient samples were syringe-filtered through a 0.2 µm filter upon arrival on shore, the filtrate was collected in a second falcon tube and the samples were immediately frozen at −20°C until further analysis. Concentrations of dissolved silicate, nitrate, nitrite and phosphate were measured with a QuAAtro nutrient analyzer (SEAL Analytical).

For the determination of in situ total particulate matter (TPM), particulate inorganic matter (PIM) and particulate organic matter (POM), part of the water from the 15 L canister was filtered in triplicates through pre-combusted and pre-weighted GF/F filters (0.7 µm, Whatman). After filtration, the filters were placed on a pre-combusted aluminum dish and dried for 24h at 60°C in a drying oven. The dry weight was measured on a microscale (Mettler-Toledo MX5) to obtain the weight of TPM, and subsequently combusted in the muffle oven at 450°C for 7 h. Finally, the samples were weighed again to obtain the weight of the remaining inorganic material on the filters. POM was calculated from the difference between the dry weight (TPM) and the combusted weight (PIM) of the material.

For the determination of Chl-a and Phaeopigments, water was filtered in triplicates through GF/F filters (0.7 µm, Whatman). To quantify the contribution of large (> 10 µm) photosynthetic cells, one sample was filtered through a 10 µm polycarbonate filter. Immediately after filtration, the filters were extracted in 100% methanol at 4°C and in dark between 12 – 24 h. Afterwards, the samples were measured with a pre-calibrated Turner Trilogy fluorometer before and after acidification with 5% HCl after (Parsons et al., 1984). In May, samples were measured with a pre-calibrated Turner AU-10 fluorometer. Chl-a/Phaeopigment ratios were calculated as an indicator for the degradation state of the algal material.

For the in situ determination of protist taxonomy (microphytoplankton and heterotrophic protists), 100 mL was filled into a brown glass bottle and fixed with a mixture of glutaraldehyde-lugol for subsequent identification and counting. Protists were identified to the lowest possible taxonomic level, verified through the World Register of Marine Species (WoRMS) and counted with an inverted light microscope (Nikon Eclipse TE-300 and Ti-S) using the Utermöhl method (Edler & Elbrächter, 2010; Utermöhl, 1958).

## 2.4 DOM sample processing

DOM extraction was performed using solid phase extraction (SPE) following the procedure of (Dittmar et al., 2008) with the addition of pre-soak of SPE sorbent with Methanol (HiPerSOLV CHROMANORM®, 99.8% VWR chemicals) 4 – 6 h prior to extraction. Fjord water samples were filtered (GF/F Whatman®, 0.7 µm, pre combusted) and acidified (pH 2 with HCl, double distilled, AnalaR® NORMAPUR®, VWR chemicals). Filtered and acidified samples (1 L) were extracted (at 15 mL/min) with a modified styrene-divinylbenzene polymer sorbent (500 mg PPL, Agilent Bond Elut™) and 0.1% v/v formic acid (HiPerSOLV CHROMANORM® VWR chemicals) was used for salt removal. Final elution was achieved using methanol (HiPerSOLV CHROMANORM®, 99.8% VWR chemicals) and stored at −20°C in the dark. SPE-DOC recovery was processed by evaporating methanol extracts and redissolving in ultrapure water (31 – 85% DOC recovery F treatment; Fig.

S9). Procedural blanks were processed using ultrapure water (HiPerSOLV CHROMANORM®, VWR chemicals) and followed the same procedure as samples.

### 2.4.1 Mass spectrometry analysis

DOM samples were analyzed by high performance liquid chromatography coupled to high resolution mass spectrometry (HPLC-HRMS). Liquid chromatography was performed using an Agilent 1100 Series system with a polar C18 column (Kinetex®, 2.1 x 150 mm, 2.6 μm bead size, 100 Å pore size) with mobile phase (A) 0.1% formic acid in LCMS grade water and (B) 0.1% formic acid in 80:20 acetonitrile : LCMS grade water (v/v). Samples were diluted in 5% v/v Acetonitrile solution (LiChrosolv, Merck) and 20 μL was injected at an initial flow rate of 150 μL min-1 with mobile phase (A) at 95% and mobile phase (B) at 5%. After 10 minutes, the Acetonitrile mobile phase (B) was increased to 95% for 2 minutes and then decreased to 5% where it was held isocratic until 15 min. Mass spectrometric analysis was completed via an LTQ-Velos-Pro Orbitrap MS (Thermo Scientific, Germany) using an electrospray ionization source (ESI) operating in negative mode (spray voltage: −3.1kV, capillary temperature: 275 °C). Blanks consisting of mobile phase A were injected periodically between samples. Each spectrum was internally calibrated in lock mass mode using three expected compounds capsaicin, fusidic acid sodium, glycyrrhizic acid ammonium salt with 304.1921, 515.3378, and 821.3965 negative m/z respectively, providing suitable accuracy and precision (< 1 ppm) in the mass range 150 – 800 m/z. Data was collected at a resolution mode of 100000. More detailed instrumentation parameters are reported elsewhere (Fonvielle et al., 2023).

### 2.4.2 DOM data processing

Mass spectrometry data were exported from the mass spectrometer and converted to mzXML files with ReAdW then processed further using MATLAB (version 2019b). A MATLAB routine was developed in-house and available with raw mzXML files. Molecular formulas were assigned between 150 – 800 Da masses. Formulas were limited with the following criteria: Carbon (4 – 50), Hydrogen (4 – 100), Oxygen (2 – 40), Nitrogen (0 – 2), Sulfur (0 – 1), H/C = 0.3 – 2.2, O/C = 0 – 1, DBE-O = –10 – 10, valence electron must be equal to an even number, and formulas which contained both nitrogen and sulfur, [13]C and nitrogen or sulfur and [13]C were removed. A mass error of 0.7 ppm was allowed for formula assignment. Peak intensities with formula assignments were normalized to sum $1 \times 10^6$ for each sample. Further descriptions for the different DOM parameters are given in Table 1. Compounds known as terrestrial peaks (t-peaks), commonly found in rivers (Medeiros et al., 2016), are identified in our experimental mass spectrometry data by matching their molecular formulas and molecular masses to those of reported t-peaks.

**Table 1:** Explanation of DOM parameters used in this study.

| DOM metric | Description | Explanation | References |
|---|---|---|---|
|  |  |  |  |

| | | | |
|---|---|---|---|
| O/C ratio | Oxygen to carbon atomic ratio | Older DOM is generally higher in oxygen content due to bacterial and photo- oxidation processes. Higher O/C values can thus indicate a decrease of DOM bioavailability. Terrestrial DOM also contains more oxygen relative to marine DOM. | (Hertkorn et al., 2013) |
| H/C ratio | Hydrogen to carbon atomic ratio | A measure for the relative hydrogen saturation. More aliphatic molecules (higher H/C) are more energy rich thus indicate higher DOM bioavailability. | (Cai & Jiao, 2023) and references therein |
| DBE | Double-bond equivalent. Double bond containing formulas. Sum of unsaturation plus rings in a molecule | Double-bonds are more difficult to break up (more energy is needed for breaking down the compound); therefore, higher DBE values typically indicate a decrease in DOM bioavailability. | (Cai & Jiao, 2023) and references therein |
| $AI_{mod}$ | Modified Aromaticity index. Describes poly-aromatic hydrocarbons. | More aromatic rings in DOM molecules lead to a higher AI and indicate low bioavailability and high recalcitrance. DOM of high AI are mostly found in deep waters. | (Koch & Dittmar, 2006a) |
| MW | Molecular weight | A measure for the size of DOM molecules which give insight into reactivity. | (Flerus et al., 2012) |
| CHO/S/N | Carbon, hydrogen, oxygen /sulfur/ nitrogen containing formulas | Heteroatoms in organic formulas such as nitrogen and sulfur contain nutrient rich components necessary for microorganisms | |


## 2.5 Statistical analyses

To visualize the seasonal biogeochemical cycle in Ramfjorden, a PCA was performed on standardized biological (POC, POC/PN, protist abundance and biomass, FCM data, Chl-a) and environmental (temperature, salinity) variables which were available for all months (except November) between September 2020 and August 2021. Subsequently, a similarity profile

routine (SIMPROF) analysis (Clarke et al., 2008) was applied to the same dataset to identify significant clusters of the sampling months (significance $\alpha$-level = 0.05) based on the biogeochemical parameters. This allowed us to divide the seasonal cycle in Ramfjorden into distinct biogeochemical periods without prior grouping of the samples. To test our main hypothesis on whether biologically active periods with higher POM concentrations have a higher potential for aggregation of DOM in comparison to the winter period, we performed a t-test on the difference (t1-t0) in experimental POC concentrations between

the two biogeochemical periods delineated by the SIMPROF test ("winter" and "productive" period).

All statistical analyses were performed with the computing environment R (Version 4.2.2 (R Core Team, 2018), MATLAB (version 2019b) and the Software Past 4 (Version 4.14, 2022 (Hammer, 2001)).


The equation for weighted average for DOM metrics (H/C$_{wa}$, O/C$_{wa}$, AI$_{mod\ wa}$, MW$_{wa}$) is shown below. Here, $I_i$ is the signal normalized intensities for a given formula and $A_i$ represents the DOM metric value for that formula and F is the total number of formulas per sample.

$$wa = \frac{\sum_{i=1}^{F} I_i \cdot A_i}{\sum_{i=1}^{F} I_i}$$

(1)

The standard deviation shown in Table 2 is calculated for the mean DOM metric in each treatment. Here, $wa_i$ is the intensity weighted average for each sample and (N) represents the total number of samples per treatment and $\overline{x}$ is the sampling mean for the treatment.

$$SD = \sqrt{\frac{1}{N-1} \sum_{i=1}^{N} (wa_i - \overline{x})^2}$$

(2)

Additionally, the weighted standard deviation (SD$_w$) for compounds in each sample is estimated by the following equation and shown in Table S2. Normalized intensities for a given formula are represented by $I_i$. DOM metric value is represented by $A_i$, and F is the total number of formulas per sample. The weighted average, $wa$, equation is the same as Equation (1).

$$SD_w = \sqrt{\frac{\sum_{i=1}^{F} I_i (A_i - wa)^2}{\sum_{i=1}^{F} I_i}}$$


(3)

The standard error of a single mean's true estimated value (and not the distribution of the population, which is estimated by SD) is approximated (as the SEM) by the following equation where the weighted standard deviation, SD$_w$, is divided by the square root of the number of identified formulas, $F$.

$$SEM = \frac{SD_w}{\sqrt{F}}$$


The following equation is used for calculating standard error of the difference of means ($SEM_{x_1-x_2}$) between treatments as shown in Table 2. Here, the standard deviations are divided by the number of samples, N, for each treatment.

$$SEM_{x_1-x_2} = \sqrt{\frac{SD_1^2}{N_1} + \frac{SD_2^2}{N_2}}$$


(5)

## 3. Results

### 3.1 The seasonal biogeochemical cycle in Ramfjorden

In winter, fluorescence was low throughout the whole water column (0.7 - 1 RFU) and density, turbidity and oxygen saturation did not show changes with depth (Fig. S3). PIM values were slightly higher in the winter months (around 0.5 µg/mL between
October - February, except for December) compared to the rest of the year (0.12 - 0.32 µg/mL; Fig. S2a1). Nutrients were increasing from November on and peaked in March, from 2.75 in November to 6 µM in March (Nitrate), 0.2 - 0.5 µM (phosphate) and 2.1 - 4.6 µM (silicate; Fig. S2g-i). By contrast, nitrite peaked in October with 0.1 µM, decreased until January (0.05 µM), and increased again until April (0.08 µM). Chla:Phaeopigments were lowest in December and January and started to increase already in March (Fig. S2w). In April, DOC, Fluorescence, turbidity, TPM, POC, Chla, protist abundance, bacterial
abundance and activity as well as nano- and picophytoplankton abundances increased sharply, along with a strong drawdown of nutrients (nitrate: 0.4 µM, phosphate: 0.1 µM, silicate: 1.5 µM; Fig. S2). Nitrite followed this trend later in May (0.03 µM; Fig. S2f). Hereafter, nutrient concentrations remained low (Nitrate < 1 µM, Phosphate < 0.3 µM) throughout the summer. In April, over 80% of the total Chl-a was > 10µm. Chl-a and POC decreased in May (0.9 mg m$^{-3}$ and 10 µM, respectively), coinciding with a sharp peak in virus abundance, but increased again in June (6.62 mg m$^{-3}$ and 35 µM, respectively) when also
bacterial abundance and activity were highest. DOC and Chla:Phaeo followed this trend. DOC peaked in August (233 µM). Another, lower peak of Chl-a and POC was observed in September (4.4 mg m$^{-3}$ and 18 µM, respectively). POC made up less than 17% of total OC during the whole year, with highest percentages in June, July and September. POC:PN ratios were highest in March (> 10) and lowest in April (~5; Fig. S2z). Between May and October, POC:PN rations remained comparable.

Most of the year, the microphytoplankton community was dominated by the diatom genus *Chaetoceros* (Fig. S2x). In April, communities were dominated by both *Chaetoceros* sp. and the prymnesiophyte *Phaeocystis pouchetii* (up to 4 x 10$^5$ cells mL$^{-1}$), in June by *Chaetoceros filiformis*, the genera *Thalassiosira* and *Pseudo-nitzschia* and in July mainly by *Pseudo-nitzschia*. Unidentified flagellates and *Chaetoceros laciniosus* dominated in August, while September was dominated by ciliates (especially *Strombidium conicum*) and various dinoflagellates that can be mixotroph. The contribution of picophytoplankton

was relatively low, with the highest abundance of *Synechococcus* sp. was observed in autumn (max 15000 cells mL$^{-1}$, Fig. S2r), while the highest abundance of picoeukaryotes was observed in April (9,000 cells mL$^{-1}$).

The PCA based on the described environmental and biological variables shows a clear seasonal pattern in Ramfjorden (Fig. 3). Along PC1, which explained 51.28% of the total variance, a clear distinction between winter - autumn - spring was present.

During winter, there were elevated levels of nutrients and POC/PN ratios, with the highest concentrations of silicate and phosphate occurring prior to the bloom in March. In April, POC levels, bacterial activity (HNA/LNA) and protist abundance reached their peak with the onset of the spring bloom. The second peak in June was characterized by highest bacterial abundance and high Chla/Phaeopigment ratios. Along the second PC axis (explaining 17.3 % of the variance), spring and winter are separated from summer and autumn (August, September, October). Here, the biological production becomes

increasingly regenerated and is characterized by highest virus abundances and bacteria of low HNA/LNA and dead phytoplankton. During this time, the water temperatures are highest (Fig. S3) and nitrite concentrations exhibit a peak (Fig. S2f). Moreover, salinity is lowest, most likely due to increased runoff from land and precipitation (Fig. S3).

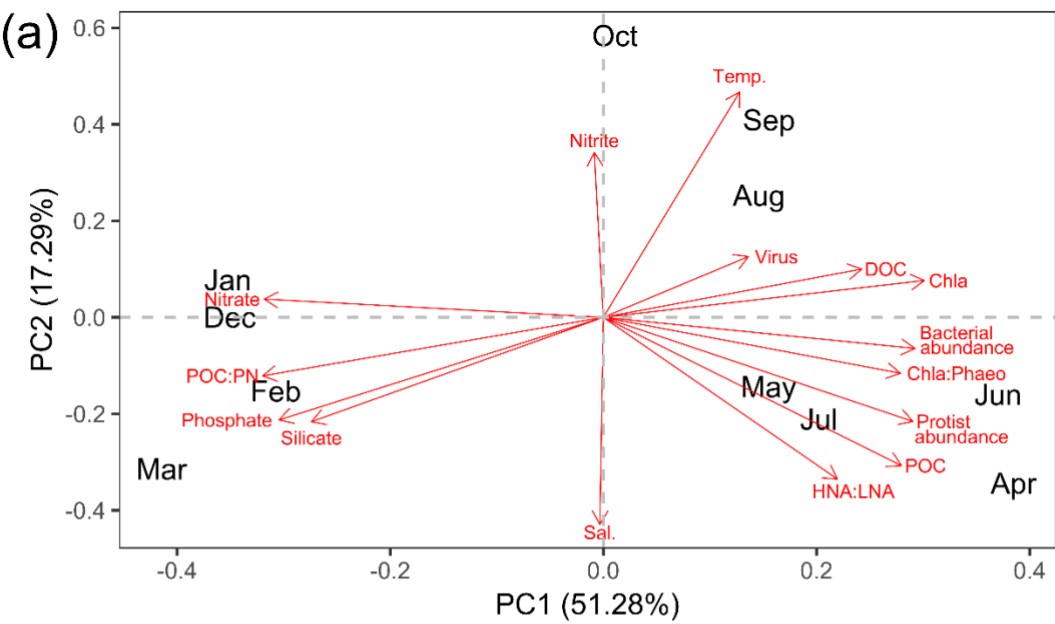

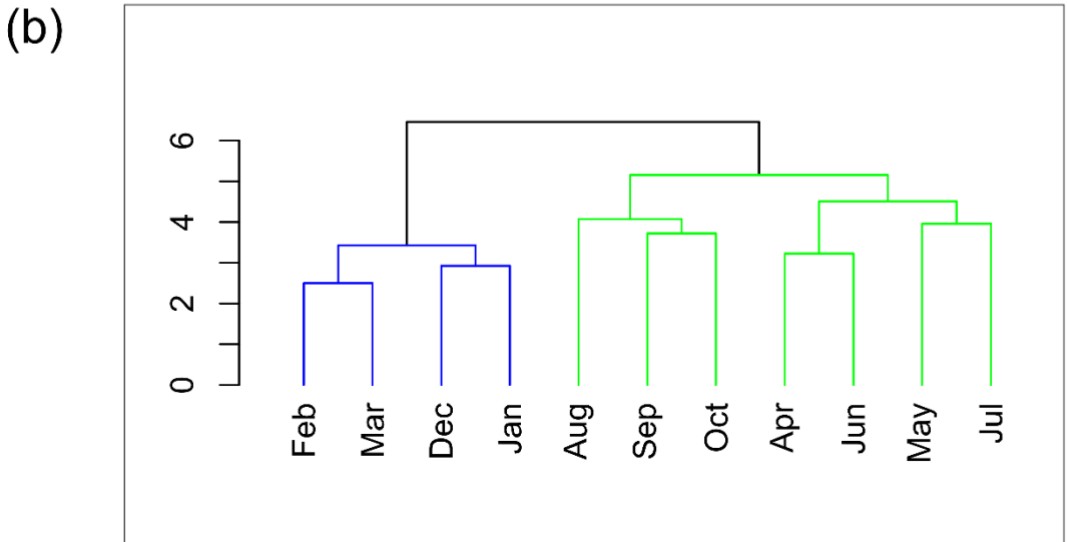

**Figure 3: The environmental and biological characterization of Ramfjorden (Tromsø, Norway) between September 2020 and August 2021.** a) Visualization of the Principal component analysis (PCA) performed on environmental and biological data (in red) collected every month between September 2020 and August 2021 in Ramfjorden. b) Visualization of the similarity profile routine analysis (SIMPROF) based on the same environmental parameters (euclidian-distance based, significance threshold alpha = 0.05).

The SIMPROF analysis in Fig. 3b based on environmental and biological data delineates two seasonal phases of the Ramfjorden ecosystem: a period of high biological activity between April - October (hereafter referred to as "productive

period"), and a period with limited light availability for phytoplankton production between November - March (hereafter referred to as "winter period"). Within the "productive period", a spring scenario (April, June, July) is separated from a summer/autumn scenario (August, September, October; however, this separation is not significant in the SIMPROF analysis). In the following, we are describing the processes in the DOM–POM continuum as observed in our experiment during the two
contrasting scenarios ("productive period" and "winter period") in the fjord environment.

## 3.2 Experimental results

### 3.2.1 Changes of  particulate organic carbon (POC)  concentrations and related parameters

The change of POC concentrations from the start (t0) to the end of the incubation (t1) of filtered (F) and unfiltered (UF) treatments followed a general similar pattern (relative increase or decrease) throughout the year (Fig. 4). In winter (December
and February), after 36 h of incubation, we measured lower POC concentrations in F water at t1 compared to t0 (a decrease of the mean POC concentration by −2.55 µM ± 0.8 (around -50% relative to t0). There was an increase of POC during the incubation period in the filtered water (F) of April, June, and September with a mean of +1.6 µM ± 0.5 (Fig. 4; April: 76% increase relative to t0; June: 88%; September: 70%). The decrease in POC in winter and the relative increase in April was followed in a similar manner in UF water (t0 is not available for UF in September). However, in June, the relative increase of
POC concentrations was 4 times higher in UF compared to F. For the F treatment, a t-test revealed significant differences in the change of POC concentrations between winter and the productive period ($p = 0.04$, t-value = 4.6, df = 3); whereas this difference was not significant for the UF treatment ($p > 0.05$, t-value = 1.6, df = 1).

Bacterial biomass in F water did not show large differences after the incubation relative to the start (t0) from September until
April, however it increased in June and August (Fig. 4b). The patterns were different in UF water, where in winter, a decrease of bacterial biomass was observed, while in spring and summer, there was an increase (Fig. S4). Despite performing the incubation in darkness there was an increase in pico-sized phytoplankton cells in the unfiltered (UF) fraction in August. At this time the system had highest concentrations of small phytoplankton ($< 2$ µm). Picoeukaryotes and *Synechococcus* increased in August from 4,000 cells at t0 to 24,000 cells mL$^{-1}$ after 36hr (Fig. S4), equalling a biomass increase of ca. 2 µM carbon
(Fig. 4c).

In F water, bacterial activity (HNA/LNA) increased relatively constant throughout the year, while in UF water, little change was observed in winter and August, while April was characterized by a sharp increase in activity (Fig. 4d). Similar to bacterial activity, EPS always increased in both treatments and throughout the whole year (sampling apart from February; Fig. 4e).
Conspicuously, while for UF water the highest increase in POC was observed in June, the highest increase in EPS was observed in April. In F water, the highest increase in EPS was present in June.

Initial (t0) DOC concentrations followed a similar pattern as the concentrations in the field (Fig. S2, S4). In September, concentrations increased in both, F and UF water (about +60 µM F and 180 µM UF; Fig. 4f). In October and December, DOC concentrations still increased in F water (+73.6 to +27.2 µM, respectively), but increased very little or even decreased in UF water. In April and June, DOC concentrations decreased in F but increased in UF water (Fig. 4f).

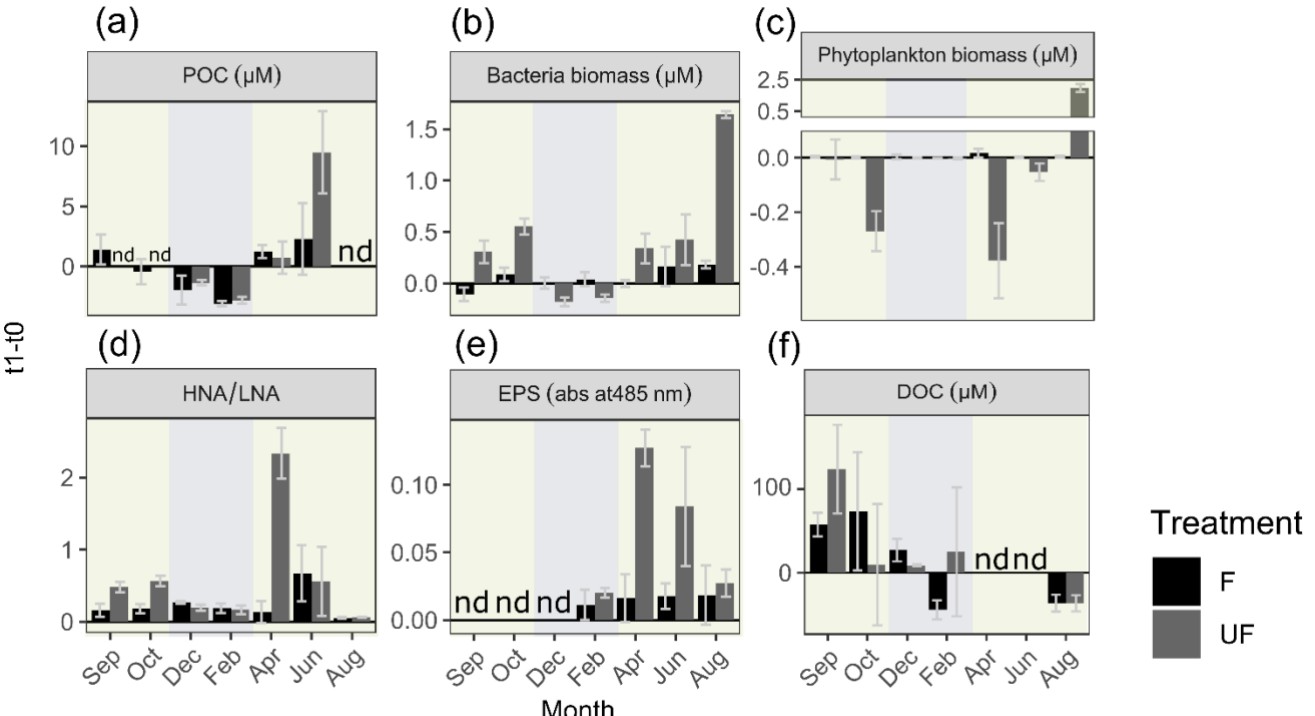

**Figure 4: Change of experimental parameters over the course of the incubation.** Change of concentrations from start relative to end of incubations (and their standard deviations; n = 2-5) of (a) particulate organic carbon (POC), (b) bacteria biomass (µM), (c) phytoplankton biomass (µM), (d) change of ratio of high nucleic to low nucleic acid bacteria (HNA/LNA) (e) change of concentrations of extracellular polymeric substances (EPS; in relative absorption at 485 nm) and (f) dissolved organic carbon (DOC) (µM) . F = filtered water, UF = unfiltered water, nd = no data. The background colors indicate the statistically identified winter (blue) and biologically productive (green) period, respectively.

### 3.2.2 DOM patterns in autumn and winter

The molecular composition of DOM was analyzed at the beginning and end of incubation experiments conducted during contrasting periods of the year. September incubations represent a productive period, whereas experiments in December and February indicate a winter period. Additionally, October incubations were also analyzed for DOM composition to assess the transitional period. Metrics for DOM composition were reported as signal intensity weighted average (wa) values of hundreds

of DOM formulas per treatment (Table 2). A table presenting the individual mean values per sample for each DOM metric is provided in the supplementary material (Table S2).

**Table 2: High resolution mass spectrometry results of DOM metrics showing intensity weighted means of hydrogen to carbon ($H/C_{wa}$), oxygen to carbon ($O/C_{wa}$), molecular weight ($MW_{wa}$) and modified aromaticity index ($AI_{mod\ wa}$).** The standard deviation (SD) is computed (Equation 2) for the start (t0) and end (t1) of incubation of the 36 hours incubation for the filtered (F) fjord water treatment (N = 3). The standard error of the difference of means ($SEM_{x_1-x_2}$) is computed (Equation 5) for each experiment.

| Treatment | $H/C_{wa}$(SD)($SEM_{x_1-x_2}$) | | $O/C_{wa}$(SD)($SEM_{x_1-x_2}$) | | $MW_{wa}$(SD) ($SEM_{x_1-x_2}$) | | $AI_{modwa}$(SD)($SEM_{x_1-x_2}$) | |
|---|---|---|---|---|---|---|---|---|
| Sep t0 | 1.31 (0.01) | | 0.49 (0.01) | | 361.40 (5) | | 0.21 (0.00) | |
| Sep t1 | 1.26 (0.01) | (0.015) | 0.52 (0.01) | (0.011) | 365.09 (3) | (5) | 0.24 (0.00) | (0.004) |
| Oct t0 | 1.27 (0.00) | | 0.51 (0.00) | | 364.53 (1) | | 0.23 (0.00) | |
| Oct t1 | 1.28 (0.01) | (0.008) | 0.51 (0.01) | (0.006) | 361.95 (2) | (1) | 0.22 (0.00) | (0.003) |
| Dec t0 | 1.28 (0.00) | | 0.51 (0.00) | | 369.83 (2) | | 0.22 (0.00) | |
| Dec t1 | 1.29 (0.01) | (0.006) | 0.51 (0.00) | (0.003) | 366.95 (1) | (2) | 0.22 (0.00) | (0.002) |
| Feb t0 | 1.28 (0.00) | | 0.51 (0.01) | | 366.05 (1) | | 0.22 (0.00) | |
| Feb t1 | 1.30 (0.00) | (0.003) | 0.50 (0.00) | (0.005) | 363.38 (3) | (0) | 0.21 (0.00) | (0.002) |

During the productive period, studied in the September incubations, average hydrogen to carbon atomic ratios ($H/C_{wa}$) decreased for both UF and F treatments (Fig. 5). Simultaneously, the average aromaticity ($AI_{mod\ wa}$) and oxygen to carbon atomic ratios ($O/C_{wa}$) increased for both F and UF treatments (Fig. 5). Average molecular weight ($MW_{wa}$) increased at the end of incubation in F treatment and decreased at the end of incubation for UF treatment. Notably, a removal of more saturated compounds ($H/C > 1.5$) is observed at the end of incubation (t1) for F (Fig. 6a) and UF (Fig. S5) treatments in September. Additionally, low SPE-DOC recoveries were observed in September and October (Fig.S9). This indicates a higher proportion of hydrophilic material (Kirchman et al., 2001; Goldberg et al., 2009) which are compounds that are not well retained by our SPE sorbent (Grasset et al., 2023).

Molecular weight patterns are shown in mass spectra (Fig. 7b) where a decrease in relative intensities of low molecular weight compounds is observed from start (t0) to end (t1) of September incubations for F treatment. These low molecular weight compounds (< 250 m/z) are highlighted on Fig. S6 and show formulas with higher relative intensities at $H/C_{wa}$ ratios greater than 1.3.

Winter months (December, February) incubations showed a contrasting pattern with an increase in hydrogen saturated formulas ($H/C_{wa}$) in both F and UF treatments (Fig. 5 a). During the same period, a decrease in average aromaticity ($AI_{mod\ wa}$) and oxygen rich formulas ($O/C_{wa}$) was observed in both F and UF treatments (Fig. 5). Molecular weight ($MW_{wa}$) decreased

during the incubation period in the F treatment and increased during UF incubation (Fig. 5). Moreover, there was a decrease of less saturated formulas (H/C < 1.5) in both F treatment (Fig. 6) and UF treatment (Fig. S5) and slight increase in intensities of more saturated formulas for F (Fig. 6) and UF treatment (Fig. S5) at the end of incubation (t1). Additionally, the highest SPE-DOC recovery, ranging from 74% to 85%, occurred in December t0 (Fig. S9).


Additionally, changes in DOM metrics for October incubation show transitional patterns from spring to winter months (Fig. 5) with the decrease in formulas with lower H/C ratios (Fig. 6b).

Molecular weight patterns are shown in mass spectra for winter incubations (Fig. 7c) where a decrease in relative intensities
of higher molecular weight compounds is observed at the end of incubation (t1). The molecular ratios of formulas higher than 570 m/z are shown in Fig. S6 composed of formulas in the mid O/C and H/C region of the van Krevelen diagram.

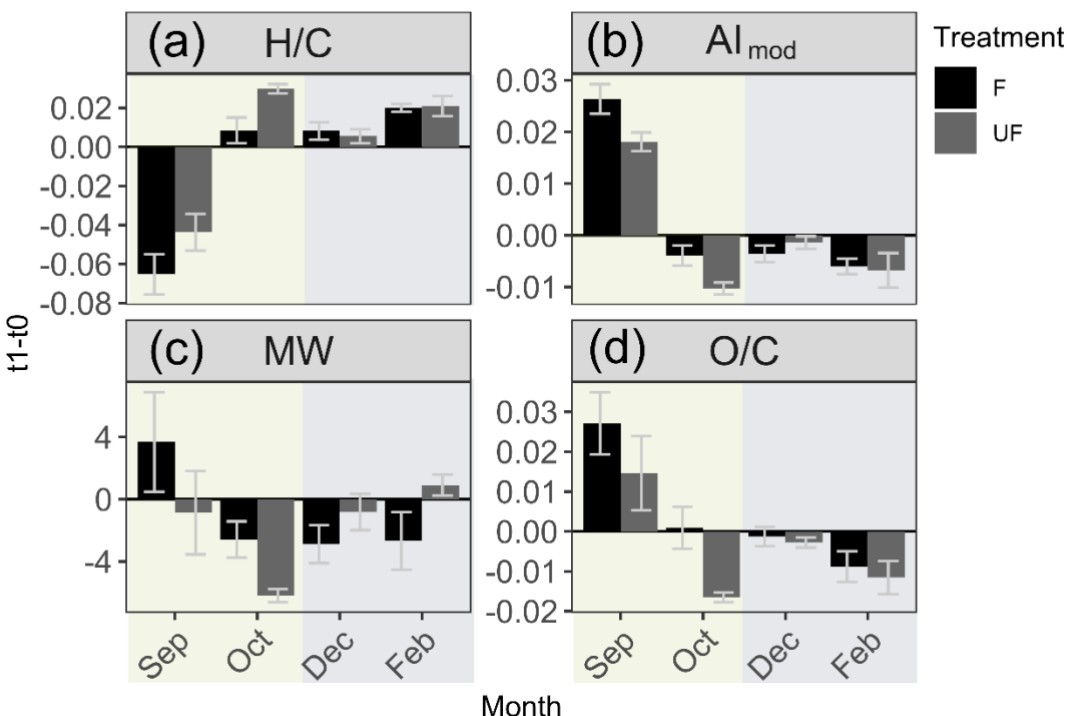

**Figure 5: Change of DOM metrics during experiment incubations.** a) change in intensity weighted average DOM metrics in seawater at t1 relative to t0. a) hydrogen to carbon ratio (H/C), b) modified aromaticity index ($AI_{mod}$), c) molecular weight (MW), d) oxygen to carbon ratio (O/C). The first treatment is filtered (F) seawater at start and end of incubation (t1 − t0) and the second treatment is unfiltered (UF) seawater for the duration of the incubation then filtered immediately prior to sampling and compared to filtered at the start (t0) incubation (t1 − t0). Error bars show the standard error of the difference of the means. The background colors indicate the statistically identified winter
(blue) and biologically productive (green) period, respectively (SIMPROF; $p < 0.05$).

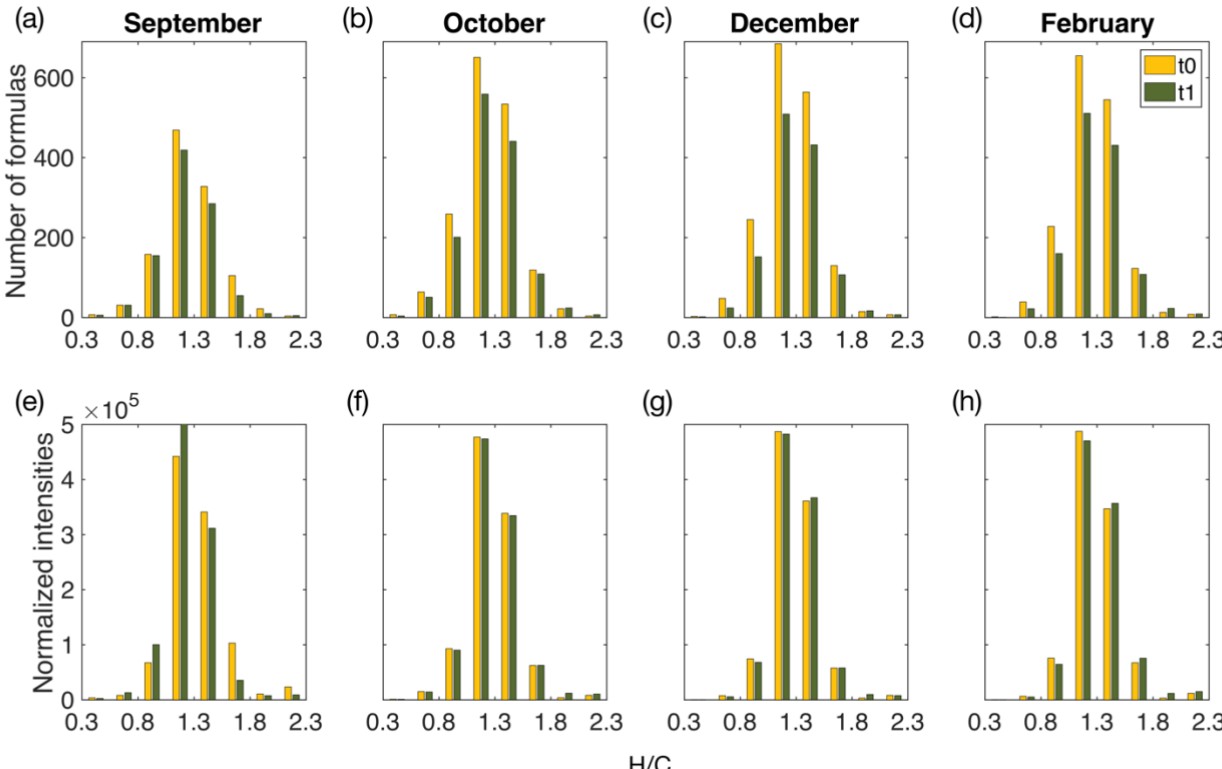

**Figure 6: Histograms of identified organic matter formulas and normalized intensities determined by high-resolution mass spectrometry during start (t0) and end (t1) of incubations of F treatment water.** Histograms of all identified molecular formulas are plotted according to the hydrogen to carbon (H/C) atomic ratio for incubation experiments in a) September, b) October c) December and d) February. Histograms of normalized intensities of identified formulas are shown according to the hydrogen to carbon (H/C) atomic ratio for incubation experiments in e) September, f) October, g) December and h) February. The start of the incubation (t0) is shown in yellow and the end of the incubation (t1) in green. F treatment refers to filtered water (0.7 µm) at the start and end of incubation (36 h). September shows a decrease in formulas and intensities with relatively higher H/C ratios whereas winter months (December, February) show decrease of formulas with low and middle range H/C ratios during incubation period. October indicates a transition period with loss of formulas across H/C ratios. Changes in intensities and formulas for UF treatment can be found in Fig.S5 in supplementary material.

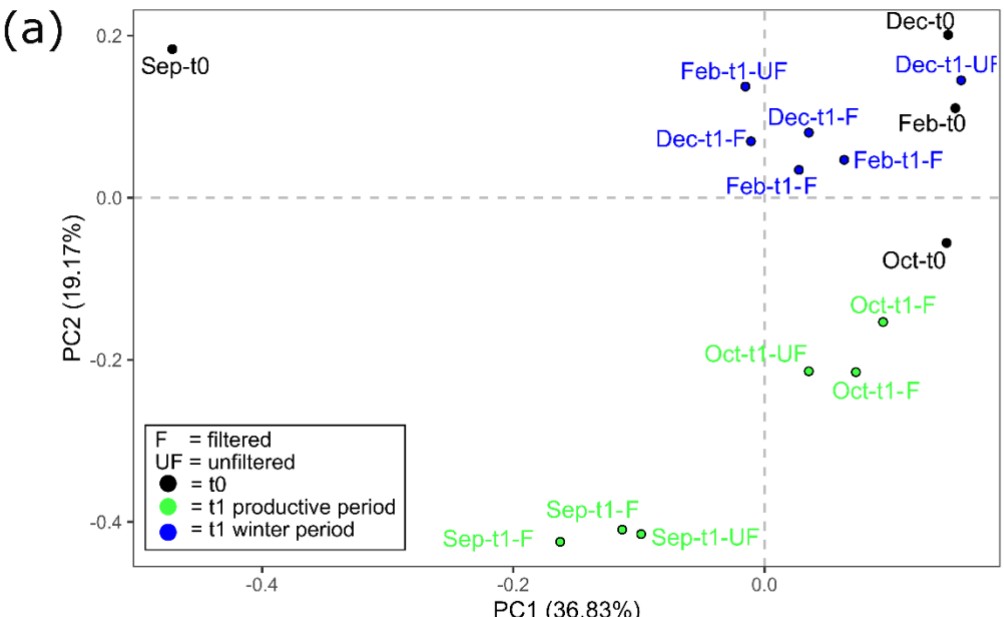

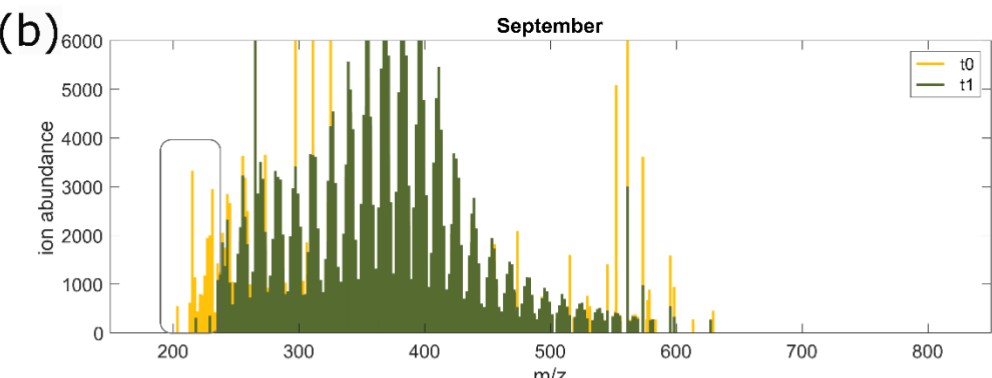

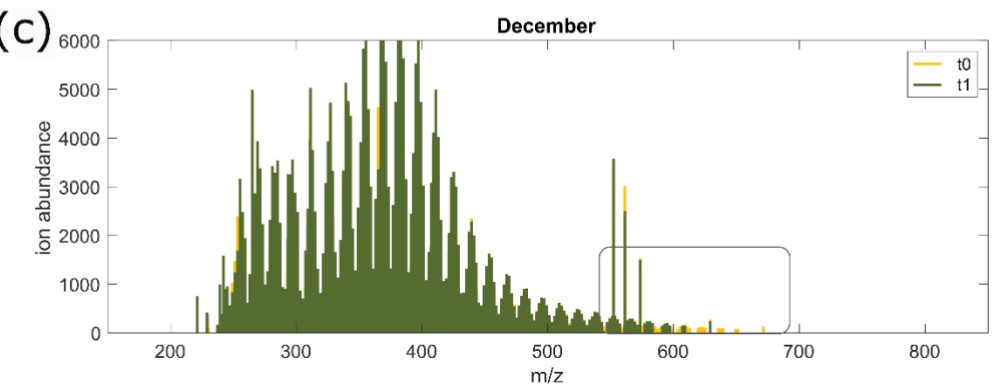

 **Figure 7: Mass spectra results from the aggregation experiment.** a) Visualization of PCA performed on DOM mass spectra from all samples from the experiment. t0 (black) and t1 samples from winter (blue) and the biologically productive period (green) are distinguished by colour. Mass spectra of filtered treatment samples from b) September at t0 (yellow) and overlay t1 (green); and c) December t0 (yellow) and overlay t1 (green). Changes between incubation periods are highlighted in the rectangles with decrease in molecular weight (m/z < 250 Da) in September from the start of incubation (t0) relative to end of incubation (t1) and decrease in high molecular weight in (m/z > 600 Da)
in December from start (t0) relative to end of incubation (t1).

## 5 Discussion

### 5.1 The seasonal biogeochemical cycle in Ramfjorden

The spring bloom in Ramfjorden and other nearby fjords is initiated at the end of March or beginning of April within a deeply mixed water column (Riebesell et al., 1995; Vonnahme et al., 2022; Walker et al., 2022). The sharp increase of DOC,
fluorescence, turbidity, TPM, POC, Chl-a, protist abundance, bacterial abundance and activity (HNA/LNA) as well as nano- and picophytoplankton abundances in April, along with a strong drawdown of nutrients demonstrates the onset of the spring bloom. Consequently, the water became turbid due to the increase of organic particles (Fig. S3, Fig. S2) and the photosynthetic activity elevated oxygen concentrations, which was consumed in the following months due to increasing heterotrophic activity (Fig. S3). Hereafter, nutrient concentrations remained low (Nitrate < 1 µM, Phosphate < 0.3 µM) throughout the summer. In
September, an autumn bloom can develop (Vonnahme et al., 2022) and high zooplankton biomass can occur (Coguiec et al., 2021). Presumably, an autumn bloom developed in September during our study as well, because POC concentrations, standing stocks of total Chl-a, contributions of large phytoplankton, mostly dinoflagellates (Chl-a > 10 µm) and ciliate abundance were similar to April and July (Fig. S2u, v, x, y).

During winter (December – March), photosynthetic activity was inhibited due to the polar night, and accordingly particulate organic matter, protist abundance and biomass as well as Chl-a concentrations were at their minimum (Fig. S2u, x). The water column is deeply mixed between December and April (Fig. S3), displaying no changes in density, turbidity and oxygen saturation with depth, and nutrients were redistributed, as demonstrated by elevated nutrient concentrations in the surface water. Despite low biological activity, bacteria continue to decompose the available algal remains during this period
(Vonnahme et al., 2022). This is also supported by the high POC/PN and low Chl-a/Phaeopigment ratios of organic particles during winter, which demonstrate that they were in a regenerated state (Fig. S2w, z).

### 5.2 DOM and POM transformations in the biologically active period

The experiments showed a decrease of POC concentrations in both filtered (F) and unfiltered (UF) water during winter (December and February), suggesting a net dissolution or degradation of particles. In contrast we measured an increase in POC
concentrations in filtered (F) water during the "productive period" (April-September). This suggests aggregation of DOM under high biological activity. Aggregation is promoted when primary production occurs, and phytoplankton exudates increase DOM concentrations in the aquatic environment as described in numerous studies (Alldredge & Jackson, 1995; Burd &

Jackson, 2009; Engel et al., 2004; He et al., 2016 and references therein; Orellana & Leck, 2015 and references therein; Passow, 2002b and references therein). The increase in POC, however, did not result in a corresponding decrease in DOC (Fig. 4), likely due to the sticky nature of extracellular polymeric substances (EPS), predominantly produced during the biologically active season (Chen et al., 2021). EPS can promote the adsorption of both DOC and POC onto filters and containers leading to complex changes in the dissolved and particulate carbon budget (see further discussion in Section 5.3.2). The increase of EPS concentrations during both the productive and the winter period, demonstrate that the behaviour of gels and colloids as precursor material play an important role in aggregation-dissolution processes. Although changes of POC concentrations in F water between the end and the beginning of the incubation were difficult to quantify due to high variability in individual tanks and the short incubation time (36h), the mean changes in the POC pool between the two contrasting seasons (winter: December, February and productive period: September, April, June) were significantly different from each other ($p = 0.04$, t-value = 4.6, df = 3). Simultaneously, DOM was characterized by more recalcitrant DOM at the end of September incubations (see section 5.2.2) with opposite trends observed during winter (section 5.3).

**5.2.1 Mechanisms of the transformations of DOM and POM**

Previous studies have shown that large (up to 5 µm) polymer gels and particulate material can be reformed quickly after they have been removed from filtered sea and river water, despite consecutive filtrations and at different filter sizes (0.2 µm, 0.4 µm or 0.7 µm; Chin et al., 1998; He et al., 2016; Kerner et al., 2003; Passow, 2000, 2002b; Sheldon et al., 1967). Even when bacteria are killed, microgels of sizes between 200 nm to 1 µm can form within 30 min, and up to 5 µm after 50 h (Chin et al., 1998; Kerner et al., 2003; Sheldon et al., 1967). The same process cannot be observed in filtered artificial seawater, but addition of dissolved carbon enhances aggregation (Gruber et al., 2006; Sheldon et al., 1967). Several studies and reviews have pointed out the ability of DOM to assemble spontaneously into polymer gels, which can aggregate to particles (Chin et al., 1998; Engel & Passow, 2001; Passow, 2000). We suggest that the same process occurs during the present study, when we observe an increase of POC after 36 h in filtered (F) water taken during the biologically productive period between April and September. Although the change in POC concentration was small (between -3.10 – +2.28 µM) and varied across individual tanks (Fig. S4; most likely due to the short incubation time), we show for the first time the contrasting seasonality of this aggregation process.

DOM concentrations are elevated through phytoplankton production, which is likely why aggregation was observed during the biologically productive period. In situ EPS concentrations, representative for large sugars (> 0.4µm) follow the same seasonal pattern as protist abundance (Fig. S2t,x). Diatoms dominated the phytoplankton community during the peak Chl-a periods in April and June (Fig. S2x). In April, the mucous colony forming prymnesiophyte *P. pouchetii* also comprised a major fraction of the phytoplankton community of Ramfjorden, and during this period EPS was elevated (0.84 relative absorption at 485 nm; Fig S2t). It was also in April that we saw the largest increase in EPS during the 36h incubation, indicating that the residues from *P. pouchetii* are especially prone to reform into EPS. In contrast, in June when only diatoms dominated (*Pseudo-*

*nitzschia* sp. and *Chaetoceros filiformis*) the experimental production of EPS was approximately 30% lower, despite peak in situ concentrations of EPS in June (1.91 relative absorption at 485 nm, Fig. S2t).

**Table 3. Percentage of particulate organic carbon (POC) contribution from aggregation of filtered (F) and unfiltered (UF) treatments relative to the in situ POC concentrations.** ΔF and ΔUF refer to the end of incubation subtracted by the start of the incubation

value for each respective treatment (F and UF).

| Month | In situ POC (μM) | ΔF POC (μM) | ΔUF POC (μM) | % F POC of in situ POC | % UF POC of in situ POC |
|---|---|---|---|---|---|
| September | 17.82 | 1.40 | n.d. | 7.88 | n.d. |
| October | 13.62 | −0.44 | n.d. | −3.23 | n.d. |
| December | 3.95 | −1.98 | −1.36 | −50.12 | −34.56 |
| February | 4.02 | −3.11 | −2.83 | −77.22 | −70.48 |
| April | 74.43 | −1.24 | 0.73 | 1.67 | 0.98 |
| June | 79.96 | 2.28 | 9.49 | 2.86 | 11.87 |

The increase of POC concentrations in F water accounted for between 1.7% (April) and 7.9% (September) of the POC concentrations measured in the field (Table 3), and although our data do not show an antagonistic relationship between POC and DOC, its source is likely aggregation from the dissolved pool. Phase shifts from DOM to POM are mainly driven by physical processes such as Brownian motion, chemical changes (e.g. ambient pH, ionic concentrations, temperature, light,

bridging with divalent cations) and/or physical stress such as turbulent shear (e.g. filtration), differential settling, surface coagulation (e.g. bubbling) or bacterial motility (Engel & Passow, 2001; He et al., 2016; Kepkay, 1994; Passow, 2000; Timko et al., 2015; Verdugo et al., 2004). During our experiment, shear was probably only introduced during the filtration and at the beginning and the end of the rotation. During the incubation, the rotation merely ensured the equal distribution of the material within the tank. Still, within a short amount of time (36 h), POC concentrations increased in F water during the biologically

productive period of the year. Other studies showed that up to 35% of particulate matter present in situ can be formed in filtered water through aggregation (Riley, 1963; Sheldon et al., 1967; Valdes Villaverde et al., 2020); however, this occurred over longer time scales (several days vs 36 h in our experiment) and by the addition of shear. Keskitalo et al. 2022 demonstrate net aggregation of POC during spring freshet in an Arctic river, and POC degradation during summer between 3-12 days. These examples indicate that next to environmental conditions, different incubation time scales probably affect DOM-POM processes

differently, and more studies are needed to disentangle these effects. For the current study, we focus on the "immediate" behaviour of the DOM-POM interactions within a short time scale compared to other experiments.

DOM–POM processes are usually driven by bacterial degradation (dissolution of POM), and physical processes (DOM aggregation to POM, adsorption of DOM to particles, or defragmentation). Our study shows that although bacterial activity increased during the incubation in both seasonal periods, microbial processes during the productive period seem to play a lesser role in the DOM–POM transition compared to physical aggregation. Other studies have similarly shown that while POC concentrations remain constant after consecutive refiltrations, bacterial abundances decrease, bacterial activity remains low and substrates that are preferred to bacterial degradation can accumulate (Engel et al., 2004; Valdes Villaverde et al., 2020). Addition of sodium azide to inhibit microbial activity does not change the coagulation behaviour of polymers, but the addition of EDTA, which disperses microgels and polymers, inhibits coagulation (Chin et al., 1998). These examples support that aggregation mainly stems from physical rather than biological transformation. Similarly, Engel & Passow (2001) and Passow (2000) show that gels > 0.4 µm in size form efficiently under shear and hydraulic stress. Overall, during the biologically productive period, a substantial fraction of particulate material can originate from the DOM pool, and changes in the DOM–POM continuum, in the direction of aggregation, have shown to be dominated by aggregation processes uncoupled from bacterial activity.

Changes in DOM composition during the incubation period for F and UF treatments were similar compared to the start of incubation (t0) and indicated little effect of larger organic size fractions (0.7 – 90 µm) on the composition of DOM during the 36h incubations (Fig. 6 & Fig. S5). However, DOM compositional changes at t1 relative to t0 were contrasting for productive versus winter period regardless of treatment, thus indicating that DOM compositional changes were driven by abiotic process and/or microbial communities in the dissolved fraction (< 0.7 µm). Due to similarities between F and UF treatments (Fig. 6 & Fig S5), we primarily focus on discussing F treatment in DOM composition sections unless otherwise stated.

### 5.2.2 Decreased DOM lability during September

Molecular composition analysis of DOM in September F treatment incubations indicates the removal of more saturated formulas and decrease in intensities of these formulas (H/C > 1.4; Fig. 6a), which indicates a decrease in overall lability of DOM compounds (D'Andrilli et al., 2015, 2023). These changes could be explained by microbial degradation and/or abiotic aggregation of the saturated compounds. Simultaneously, an increase in average aromaticity ($AI_{mod\ wa}$) of DOM compounds was observed during incubation and is likely due to the removal of saturated molecules which tend to be low in aromaticity (Koch & Dittmar, 2006a). Additionally, the removal of low molecular weight compounds (< 250 Da) was also observed in mass spectra at start of incubation (t0) versus end of incubation (t1; Fig. 7). These low molecular weight compounds are mainly composed of higher H/C saturation (H/C > 1.3; Fig. S6); thus, removal of these compounds could explain the lower average $H/C_{wa}$ ratios observed. Microbial degradation of these compounds (< 250 Da) is contrary to the size reactivity continuum which proposes higher reactivity of DOM as molecular weight increases (Benner & Amon, 2015).

Previous work has shown the formation of POM via adsorption of hydrophilic and low aromatic DOM (Einarsdóttir et al., 2020). The decrease in the average H/C saturation and increase in aromaticity of DOM compounds observed during September incubations could be due to the adsorption of these compounds to POM. However, aggregation of highly saturated DOM is typically observed for larger size fractions, such as polysaccharides (Passow, 2000; Passow et al., 1994), which are outside of our mass spectrometry analysis window. These low molecular weight compounds could point to DOM precursors molecules

of the larger hydrophilic POM compounds (Orellana and Leck, 2015; Verdugo, 2004). Low SPE-DOC % recovery observed during September incubations (Fig. S9) supports the increase in these hydrophilic fractions. Although these dissolved fractions were not extracted in our DOM method, their presence could contribute to increases in POC concentrations as observed during the biologically productive period.

Average oxygenation of molecules ($O/C_{wa}$) also increased during the incubations in September. This could be due to the removal of low O/C (< 0.5) and high H/C ratio (> 1.5) formulas as seen in Fig. S8 instead of the production of highly oxygenated compounds. The preferential biological degradation of compounds with low oxygen numbers compared to oxygen rich compounds was also observed by Riedel et al. (2016). Additionally, Maie et al. (2008) have shown aggregation of highly oxygenated tannin compounds, however our experimental results did not show an increase of highly oxygenated compounds

likely due to a limited tannin source in this region as shown by the low seasonality in the tannin region (Fig. S8).

### 5.2.3 Enhanced aggregation under post-bloom conditions

POC aggregation during the experiment was higher in June compared to April (increase by 1.2 µM POC in April vs. 2.6 µM in June for F, and an increase of 0.7 µM in April vs. 9.5 µM in June for UF water). Higher aggregation under post-bloom conditions in June compared to April were not surprising, as EPS accumulate under increasing nutrient limitation and/or with

increasing concentrations of senescent cells, as it is the case during post-bloom conditions in summer (Engel, 2000; Hellebust, 1965; Mague et al., 1980; Myklestad, 1995; Passow, 2002b; Riebesell et al., 1995; Thornton, 2002). EPS are released already in the growth phase of phytoplankton during the initiation of a bloom; however, most EPS are probably present in the form of low molecular weight (LMW) compounds in the beginning of a spring bloom (Paulsen et al., 2018). This is supported by higher field DOC concentrations in April than in June, while field concentrations of EPS, which are representative for large

polysaccharides (> 0.4 µm), were lower in April compared to June (Fig. S2t). This might also explain why in April both treatments, F and UF, had a similar increase of POC levels relative to start of incubation (increase of 1.24 and 0.73 µM; respectively), although initial POC concentrations at the start of incubation (t0) were 11 times higher in UF than in F water. Despite particle concentrations being high, little precursor material was probably present in UF water to promote aggregation in the beginning of a bloom compared to a post-bloom scenario.


Later, in June, we observed a 9 times higher aggregation potential of UF water compared to April, while we only observed a 2  times higher aggregation potential of F water (Fig. 4a). In summer, towards the end of a bloom, not only does the field

concentration of EPS increase, but there is also an increase in concentration of suspended particles as shown by an increase of POC and TPM in the field (Fig. S2y,a1). Because coagulation becomes more likely when a critical concentration of particles is reached, aggregation is usually enhanced under post-bloom conditions (Burd & Jackson, 2009; Dam & Drapeau, 1995; Mague et al., 1980; Passow, 2002b; Thornton, 2014). Colloids can be scavenged by larger particles (Druffel et al., 1992; Kepkay, 1994) and can even lead to enhanced carbon sedimentation (Forest et al., 2013; Riebesell et al., 1995). Higher POC and EPS concentrations in the field during June likely increased the chance of organic matter to coagulate in UF water. Interestingly, however, experimental POC concentrations in UF water were higher in April compared to June. This might indicate that the concentration of precursor material plays a more critical role for aggregation than particle concentration. Our experiment suggests a higher contribution of DOM to the POM pool during summer compared to spring, and a higher aggregation potential in general (as seen in UF water) during a post-bloom compared to a peak-bloom scenario. It should be noted, however, that experimental EPS concentrations at the start of incubation (t0) in UF water were consistently an order of magnitude lower than the respective field EPS concentrations, which indicates that a large fraction of EPS was retained on the 90µm sieve and therefore not included in the experiment, probably because of their sticky nature. Therefore, we likely did not adequately replicate aggregation in truly unfiltered water, and our measurements can be considered as conservative estimates for aggregation in unfiltered water.

**5.3 DOM and POM transformations in winter**

Our experiment shows a decrease in POC concentrations in F and UF water after incubation during winter (December and February). In the ocean, particles are usually remineralized in the pelagic zone due to solubilization by bacteria, sloppy feeding by zooplankton or fragmentation (Iversen, 2023; Kiørboe, 2001; Svensen & Vernet, 2016). Photo-dissolution can also turn POM into dissolved material, especially in surface waters under direct light influence (Pisani et al., 2011). Since large grazers were filtered out and the experiments were carried out in the dark, we propose that bacterial degradation was responsible for the observed dissolution patterns of organic matter that was present in the winter water. This is also supported by the increase of bacterial activity in the experiment at t1 relative to the start of incubations (t0) throughout the year, and by the fact that a decrease of POC ("particle dissolution") was observed in F and UF water.

Other aggregation experiments conducted in temperate regions where light is still available during winter show that particles form from filtered water during this time of the year (Riley, 1963; Sheldon et al., 1967). Our study, however, measured a decrease in POC concentrations during winter and changes in DOM composition (see Section 5.3.1) at the end of incubation (t1) compared to the start of incubation (t0) which could indicate that this dissolution process occurs in the Arctic polar night, but not in temperate regions. However, we also measured an increase in experimental EPS concentrations in the F water during February. This is supported by the low SPE-DOC % recovery observed in February (Fig.S9). As a result, several processes might occur simultaneously within the DOM-POM continuum throughout the winter. Our results indicate that dissolved EPS molecules (< 0.45 µm) are aggregating, while at the same time POM is degraded by bacteria. Similarly, (Xu & Guo, 2018)

showed that DOM compounds in a certain size range aggregated simultaneously while other DOM compounds were degraded by bacteria.

Experimental POC concentrations at the start of incubation (4.7 – 5.3 µM in F, and 5.6 – 6.0 in UF water; Fig. S4) were similar to field POC concentrations in winter (around 4 µM, Fig. S2y), which suggests that particles were of extremely low abundance and size during the winter period (Table S1). Throughout the whole sampling period (September-August), experimental POC concentrations at t0 in UF water largely followed a similar seasonal pattern as field POC concentrations, although at lower levels. Note that the water collected for the UF treatment was sieved through a 90µm mesh, while the water collected for field measurements was not sieved. However, experimental POC concentrations at t0 in F water seemingly followed opposite patterns, with low concentrations in September (2 µM) increasing in winter until February (5.3 µM) and sharply declining in April (1.6 µM; Fig S4). A possible explanation is that during filtration of water with high particle and EPS abundance, as is the case in the productive period, small molecules are retained on the filter because they get trapped in the sticky matrix or aggregates, therefore don't pass the filter pores and lead to lower POC concentrations in F water at t0 during the productive period compared to higher POC concentrations at t0 in the winter period.

### 5.3.1 Increased DOM lability during winter incubations

DOM molecular composition analysis during winter incubations (December and February) indicates a decrease in unsaturated DOM (Fig. 6) and an increase in relative intensities of more saturated DOM (H/C; Fig. S8). Moreover, average aromaticity ($AI_{mod\ wa}$) decreased during this period, which could be due to the removal of low H/C DOM which is typically of higher aromaticity (Koch & Dittmar, 2006b). These observations indicate an increase in average lability of DOM and could be partially due to the dissolution of organic particles observed during this period as it can lead to the production of more labile DOM. This is supported by the reduction of SPE-DOC % recovery observed during December incubations (Fig S9) which indicates hydrophilic material at the end of incubation. Additionally, there was a significant decrease of a group of low H/C compounds referred to as 'terrestrial peaks' (t-Peaks) in February (t-test, p = 0.04, t-value = 2.96, df = 4) and October (t-test, p = 0.04, t-value = 3.02, df = 4). T-Peaks are a group of compounds that are commonly present in vastly different rivers as reported by Medeiros et al. (2016; Fig. S7). Removal of these compounds could contribute to the increase in average $H/C_{wa}$ ratios observed in February and October incubations. This suggests a potential degradation of t-Peak compounds during February and October, in contrast to September (t-test, p = 0.08, t-value = -2.31, df = 4) and December (t-test, p = 0.09, t-value = 2.21, df = 4) when t-Peaks did not significantly change during the incubation (Fig. S7). Arctic winter microbial communities differ substantially from spring, summer and autumn communities (Marquardt et al., 2016; Vonnahme et al., 2022; Wietz et al., 2021), and therefore the winter community is likely better adapted to degrade various carbon sources, while summer communities are specialized in degrading phytoplankton derived DOM as shown in Wilson et al (2017). These findings support that terrestrial DOM is an important carbon source for bacteria during less productive periods in Ramfjorden (Vonnahme et al., 2022), which has implications for the fate of riverine DOM in sub-Arctic fjords and a freshening Arctic Ocean.

Additionally, the winter presents a period where heterotrophs have less competition from photoautotrophs for essential
nutrients which could promote degradation of seemingly recalcitrant DOM (Dittmar, 2015).

There was also a removal of relatively higher molecular weight compounds (570 – 700 Da) as shown in mass spectra (Fig. 7c). These higher molecular weight compounds were in the mid O/C and H/C ratio region of the van Krevelen diagram (Fig.
S6) which are typically rich in carboxyl groups (Broek et al., 2020; Hertkorn et al., 2006). We suggest that the decrease of these compounds could indicate colloid formation through adsorption of higher molecular weight (> 700 Da) as it has been previously observed for carboxyl rich organic matter (Chin et al., 1998). Notably, experimental POC concentrations during the incubations indicate a dissolution, and not an aggregation of particles during the winter. This could be due to the low relative intensity of these higher molecular weight compounds (570 – 700 Da), thus no detection of increased POC
concentrations during this period.

## 5.3.2 Carbon budget

Throughout the experiment, we expected that DOC and POC would behave antagonistically, and thus an increase in POC would see a corresponding decrease in DOC, resulting in a stable amount of total organic carbon (TOC; sum of DOC and POC). However, this was not always the case during our experiments; the most pronounced change in the carbon budget was
an increase in both DOC (+180 µM UF treatment) and POC (+1 µM) in September. This suggests that our experiment was either not a closed system, or that there was an interchange with the inorganic C pool. Other studies also report synergistic relationships between POC and DOC during summer incubations and antagonistic relationships during freshet incubations (Keskitalo et al., 2022). This indicates a seasonal component to POC-DOC dynamics. Additionally, other incubation studies report large variability in changes in POC and DOC concentrations (Shakil et al., 2022). These findings highlight the challenges
in organic carbon measurements as reported elsewhere (Gardner et al., 2003; Chow et al., 2022) and are likely due to the complex dynamics between  POC and DOC such as sticky nature of EPS material which can be adsorbed onto experimental containers (Chen et al., 2021). Notably, experimental DOC values of ultra-pure water blanks show average DOC concentrations of 31 µM ± 13 (mean ± SD) and DOM molecular analysis of ultrapure water experimental blanks reveal relatively low number of contamination peaks (average number of peaks: 123 ± 33, mean ± SD), thus not affecting DOM
characterization results and ruling out contamination as a source of significant carbon addition. Moreover, the DOC concentrations measured in the experiment are well within the range of the in situ measurements (ranging from 87-233 µM; Fig. S2k).

In winter months the carbon budget had a net-loss of TOC (POC + DOC) as observed in February, which was most likely due
to bacterial respiration of OM to $CO_2$. In both October and December, a decrease in POC (-0.4 µM and -2 µM, respectively) did reflect an increase in DOC (+74 µM and +27 µM, respectively), however indicating a substantial additional source of

DOC. The high increase in TOC observed in September, October and December could be partly explained by carbon fixation in the dark by phytoplankton, as Vonnahme et al. (2022) and Hoppe et al. (2024) describe low, but measurable primary production during winter. This may play a role in the increase in carbon observed in the unfiltered treatments where phytoplankton was abundant. In the filtered treatments we did however observe a rapid increase in the activity of bacteria, presumably as in this fraction they were released of grazing pressure from bactivorous protists. Further, viruses are present in the filtered treatment and at their peak abundance (up to $1.3 \times 10^7$) in autumn. Viruses have a significant role in controlling microbial population (Suttle, 2007), which during the lytic stage can lead to a substantial production of DOM through viral lysis of microorganisms (Chen et al., 2022). We suggest that the production of DOC observed in September and October in the filtered treatment may be due to viral lysis of the otherwise rapidly growing bacterial community (relieved from grazing pressure and not limited by carbon nor nutrients). In the unfiltered treatment the presence of phytoplankton (new production) may explain the increase in DOC.

**6 Conclusions and Outlook**

With its sharp seasonal gradients, the Arctic presents a suitable place to study the influence of different environmental factors on DOM and POM dynamics. Our results confirm our initial hypothesis; that the aggregation potential of POM is higher in the productive period (an average increase of POC of $+1.6\,\mu M \pm 0.5$ within 36hr) compared to the winter period where there was an average POC decrease of $-2.55\,\mu M \pm 0.8$. Our findings also reveal that the molecular characteristics of DOM during short time frame incubations are influenced by contrasting seasonal conditions at high latitudes, with distinct transformations occurring during the biologically productive and the winter period. It is important to note, however, that the POC and DOC pool did not behave antagonistically during our seasonal incubations, similar to findings in other studies. This underlines the complexity of OM transitions with colloids, gels and biological communities likely playing an important role in these transformations, and more studies are needed to disclose the underlying mechanisms.

In winter (December and February), we observed an increase in average DOM lability that may be partially attributed to the solubilization of particles and colloids, supported by the dissolution of POC. However, the decrease in POC concentrations was much smaller than the observed changes in DOC suggesting that the changes in DOM characterization are more likely driven by microbial degradation of DOM and colloids. The microbial communities may play a crucial role in driving these distinct processes, as competition from autotrophs for essential nutrients is reduced during this period (Dittmar, 2015 and references therein). Additionally, during winter, there was a decrease in DOM compounds within the 570 - 700 Da molecular weight range, in the mid O/C and H/C region of the van Krevelen diagram, indicating the removal of carboxyl-rich compounds. This has implications for carbon cycling through the removal of recalcitrant DOM components. Winter incubations also showed a significant decrease in so-called terrestrial peaks with implications for removal of river DOM. Dissolution,

aggregation and microbial exchange of organic matter are not unique to the Arctic, but universal to all aquatic systems. With its seasonal extremes, our chosen study site provides a unique insight into the switching between dominant mechanisms.

During the "productive period" (April-September) we observed an increase of POC concentrations, which is likely attributed to aggregation processes of the colloid and dissolved pool. DOM characterization during incubations in this period showed reduced lability, suggesting microbial degradation of bioavailable DOM. October incubations showed a transitional period between productive and winter months with a decrease in labile DOM and increase in recalcitrant compounds. These findings

underline that changes during short time frames in the POM and DOM pools are highly subject to seasonal transformations and that biotic and abiotic processes drive these changes. A substantial fraction of POM can potentially originate from the DOM pool through aggregation, although the POC and DOC concentration dynamics indicate more complex processes taking place within the system. Additionally, aggregation within short time frames in the productive period (within 36 h as we have shown) demonstrate that standard POC measurements in the field under periods of high production likely underestimate actual

in situ POC concentrations, as they do not account for the dynamic exchange with the DOM and colloidal pool.

**Data availability**

All data are available on the NIRD research data archive (https://archive.sigma2.no). Field data are found under the following DOIs: 10.11582/2024.00096 (CDOM), 10.11582/2024.00116 (POC/PN), 10.11582/2024.00117 (Chl-a), 10.11582/2024.00118 (TPM), 10.11582/2024.00119 (EPS), 10.11582/2024.00120 (nutrients), 10.11582/2024.00131 (protists)

and 10.11582/2024.00132 (FCM). Experimental data can be found under 10.11582/2024.00121 (POC, FCM and EPS) and 10.11582/2024.00098 (DOC and TDN). Mass spectrometry raw files (mzXML) and MATLAB code for the DOM data can be found under 10.5281/zenodo.14272939.

**Author contributions**

MGD and YVB equally lead the study design, field work, data analysis and writing of the article. MLP and MAA contributed

to the study design and sample analysis. MAA took part in field work and was responsible for the sampling and analysis of the field parameters. MGD was responsible for the dissolved, and YVB for the particulate and biological parameters of the experiment. MLP analyzed FCM and cDOM samples. JAH contributed to the sample and data analysis of DOM, and UD analyzed EPS samples. SGK contributed to field work and sample analysis. The draft of the manuscript was written by YVB, MGD and MLP, and all authors contributed to the interpretation of the data and commented on the manuscript. All authors

read and approved the final manuscript.

**Competing interests**

The authors declare no conflict of interest.

**Acknowledgements**

We would like to thank the crew of R/V Hyas who made it possible to collect samples every month. We thank Reidar Kaasa
and Per Gjerp for manufacturing the experimental equipment and Evald Nordli for logistics. We thank Camilla Svensen, Lena
Seuthe, Rolf Gradinger and Tobias Vonnahme for their input regarding the experimental setup and fieldwork, and Tobias
Kielland, Lena Seuthe, Elisabeth Halvorsen and Anna Miettinen for help in the field and in the lab. We also would like to
thank Anna Maria Dąbrowska for the taxonomic identification of protists. We thank Murat Ardelan and Morten Iversen for
the valuable scientific discussions. This study was funded by the Research Council of Norway (#276730) through the Nansen
Legacy project (MGD, YVB, MAA, SGK, OM, MR). The work was also funded by the Norwegian university of science and
technology (NTNU; MGD, SGK) and The Arctic University of Norway (UiT; YB, MAA).

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
