# Peer review of "Contrasting seasonal patterns in particle aggregation and DOM transformation in a sub-Arctic fjord"

_EGUsphere, 2024_

## Author Comment (AC1)

**Table 2: High resolution mass spectrometry results of DOM variables showing mean intensity weighted average of hydrogen to carbon (H/C$_{wa}$), oxygen to carbon (O/C$_{wa}$), molecular weight (MW$_{wa}$) and modified aromaticity index (AI$_{mod\ wa}$).** SEM= standard error of difference between means. N = 3 for the start of incubation (t0) and end of incubation (t1) (36 hours) for filtered (F) water treatment.

| Treatment | H/C$_{wa}$ (SEM) | O/C$_{wa}$ (SEM) | MW$_{wa}$ (SEM) | AI$_{mod\ wa}$ (SEM) |
|---|---|---|---|---|
| Sep t0 | 1.31 (0.007) | 0.49 (0.006) | 361.40 (3) | 0.21 (0.002) |
| Sep t1 | 1.25 (0.007) | 0.52 (0.005) | 365.09 (2) | 0.24 (0.002) |
| Oct t0 | 1.27 (0.002) | 0.51 (0.001) | 364.53 (0) | 0.23 (0.001) |
| Oct t1 | 1.27 (0.006) | 0.51 (0.005) | 361.95 (1) | 0.22 (0.002) |
| Dec t0 | 1.28 (0.002) | 0.51 (0.001) | 369.83 (1) | 0.22 (0.001) |
| Dec t1 | 1.29 (0.004) | 0.51 (0.002) | 366.95 (0) | 0.22 (0.001) |
| Feb t0 | 1.28 (0.001) | 0.51 (0.004) | 366.05 (0) | 0.22 (0.001) |
| Feb t1 | 1.30 (0.002) | 0.50 (0.002) | 363.38 (2) | 0.21 (0.001) |

[Figure]

**Figure 5: Change of DOM parameters during experiment incubations.** a) change in intensity weighted average DOM parameters in seawater at t1 relative to t0. a) molecular weight (MW), b) oxygen to carbon ratio (O/C), c) hydrogen to carbon ratio (H/C), d) modified aromaticity index (AI$_{mod}$). The first treatment is filtered (F) seawater at start and end of incubation (t1 − t0) and the second treatment is unfiltered (UF) seawater for the duration of the incubation then filtered immediately prior to sampling and compared to filtered at the start (t0) incubation (t1 − t0). The background colors indicate the statistically identified winter (blue) and biologically productive (green) period, respectively.

[Figure]

**Boxplot of SPE-DOC % recovery:** This figure shows the SPE-DOC % recovery for the treatments at the start (t0) and end of incubation (t1).

---

## Author Comment (AC2)

[Figure]

**Figure S7: van Krevelen diagram of t-Peaks during experiment months. Each point is an identified molecular formula and plotted according to the oxygen to carbon (O/C) and hydrogen to carbon (H/C) atomic ratio of the molecular formula.** a) start of incubation (t0) are shown in black and t-Peak molecular formulas present are highlighted in orange. b) end of incubation (t1) (F treatment) are shown in black and t-Peak molecular formulas matches highlighted are in orange. T-peak molecular formulas are obtained from Medeiros et al. (2016). Results show significant decrease of t-Peaks in December incubations (t-test, $p<0.05$) and not significant in September, October and February incubations (t-test, $p>0.05$).

[Figure]

**Figure 5: Change of DOM parameters during experiment incubations.** a) change in intensity weighted average DOM parameters in seawater at t1 relative to t0. a) molecular weight (MW), b) oxygen to carbon ratio (O/C), c) hydrogen to carbon ratio (H/C), d) modified aromaticity index ($AI_{mod}$). The first treatment is filtered (F) seawater at start and end of incubation (t1 − t0) and the second treatment is unfiltered (UF) seawater for the duration of the incubation then filtered immediately prior to sampling and compared to filtered at the start (t0) incubation (t1 − t0). The background colors indicate the statistically identified winter (blue) and biologically productive (green) period, respectively.

[Figure]

**Figure S5: Histograms of all identified organic matter formulas and normalized intensities determined by high-resolution mass spectrometry during start (t0) and end (t1) of incubations of UF treatment water.** Number of identified molecular formulas are plotted according to the hydrogen to carbon (H/C) atomic ratio for incubation experiments in a) September, b) October c) December and d) February. Normalized intensities are plotted according to H/C ratios for (e-h) for respective months. The start of the incubation (t0) is shown in yellow and the end of the incubation (t1) in green for UF treatment.

---

## Author Response (AR1)

Reviewer 1:

1. The authors attempted to interpret their experimental results in the context of POM and DOM transition, arguing that DOM aggregation and POM dissolution determine the fate of these organic matters. However, the results of the incubation experiments are not strong enough to support this interpretation because: The trends of changes in POC and DOC concentrations during the incubation seem to be complex and they are not always having opposite trend. The authors also mentioned this in the section 5.3.2. The magnitudes of changes in POC and DOC concentrations are significantly different (for example,~ 2 μM changes in POC vs >200 μM changes in DOC  in Sep incubation), making it difficult to compare these changes to draw conclusions about DOM aggregation and POM dissolution.

We appreciate Reviewer #1's interest in our study and their valuable comments. We fully acknowledge the mentioned limitations of our study and have addressed them openly, as they reflect inherent technical challenges faced in all studies on the DOM-POM transition. We employ a wide range of techniques to give as many insights into organic matter continuum on a seasonal scale. These include DOM characterization (HPLC-HRMS), colorimetric determination of exopolymeric substances, biological parameters as well as DOC/POC/PN concentration. Our study was meticulously designed to be ultra-clean, free from organic matter contamination and reflected in the POC, DOC concentration and DOM characterization blanks (Section 5.3.2, Figure S1).

We are not able to quantitatively determine the flux between DOC and POC likely due to the sticky nature of extracellular polymeric substances (Chen et al., 2021), which can cause DOC and POC to be adsorbed on the experimental containers and filters leading to complex changes in the dissolved and particulate carbon budget (Vades-Villaverde et al., 2020). Additionally, the magnitude differences between POC and DOC, as mentioned by the reviewer, meant that we could not infer that the decrease in POC is directly related to the increase in DOC, however we could infer that an increase in POC (a relatively smaller fraction) may be due to the aggregation of DOC. Further, we highlight the importance of determining accurate POC and DOC measurements, which is often a challenge (Chow et al., 2022; Gardner er al., 2003; Wangersky, P., 1993), and we recommend a high number of replicates (5-10).

Due to the mentioned limitation, we focused the revised manuscript on the contrast between winter and the productive period in POC and the characterization of DOM, rather than on the quantitative differences between DOC and POC. Namely, we highlight key findings on seasonal contrasts, such as POC increases during biologically productive months and its dissolution in winter (t-test, p = 0.04). Additionally, we explore the contrasting trends in DOM lability, which increases during the winter and decreases during the productive period (Figure 5). These individual findings have implications on the DOM-POM continuum on a seasonal scale.

Related changes on the manuscript:

L11: Abstract: we rephrased the abstract highlighting the contrasts between winter and the productive period and mentioned that the changes in the OC pool were not antagonistic between DOC and POC.

L389, 428, 524, 668: We removed the terms "aggregation" and "dissolution" from the sub-headers, to highlight dynamic transformations in the OC pool, and these terms were rather part of the discussion

L422 & L183: During a recent review of the data prior to publishing to public datasets, we discovered that the t0 values for September had been incorrectly recorded. After correcting the issue, the updated t0 values are shown in the Figure 4 with no changes on the findings of the manuscript.

L525: Instead of directly implying aggregation/dissolution processes, we wrote: "The experiments showed a decrease of POC concentrations in both filtered (F) and unfiltered (UF) water during winter (December and February), suggesting a net dissolution or degradation of particles. In contrast we measured an increase of in POC concentrations in filtered (F) water during the "productive period" (April-September)"

L531: We highlighted that the changes in POC and DOC were nevertheless not antagonistic and discuss possible causes for this.

L579: We discuss the effect of different incubation times on DOC-POC transformations, citing one of the suggested references.

L751: Increase in UF treatment DOC (180 µM) during September incubations and could be due to carbon fixation in the dark as shown by Vonnahme et al. (2022) and Hoppe et al. (2024).

L755: Increase in F treatment in DOC is likely due to rapid increase in bacterial activity as they were released of grazing pressure and viral lysis of bacterial community.

L730: Section 5.3.2 carbon budget includes more studies on the shared technical challenges in DOC and POC studies.

L736: added discussion about inherent challenges in DOC and POC measurements : "Other studies also report synergistic relationships between POC and DOC during summer incubations and antagonistic relationships during freshet incubations (Keskitalo et al., 2022). This indicates a seasonal component to POC-DOC dynamics. Additionally, other incubation studies report large variability in changes in POC and DOC concentrations (Shakil et al., 2022). These findings highlight the challenges in organic carbon measurements as reported elsewhere (Gardner et al., 2003; Chow et al., 2022) and are likely due to the complex dynamics between POC and DOC such as sticky nature of EPS material which can be adsorbed onto experimental containers (Chen et al., 2021)."

Conclusions: The conclusions were rewritten to reflect more the findings of the study in line with the comment of the reviewer.

2. The authors also tried to compare the changes in DOM compositions before and after incubation. However, the analysis is questionable because the changes in metrics such as H/C, O/C, MW, and AI in Table 2 are small compared to the standard deviations. Most of the discussions are based on these changes, which are smaller than the errors. While the authors argue that these small changes are significant, they do not provide any statistical test reports

to support this argument. The authors should conduct statistical analysis to support that those small changes are still significant.

We appreciate the reviewer's comment and would like to clarify that the standard deviations originally shown in Table 2 are computed from all of the compounds in sample treatment. Based on this comment from the reviewers, we think it is more relevant for Table 2 to include the standard deviations for the replicate means rather than from the compounds. The new standard deviations on Table 2 show the low variability between replicates. We also include the standard error of the difference between the means of the incubations and appear as error bars on Figure 5. Additionally, we added Table S2 to the supplementary which includes all of the DOM metric values with the standard deviations and standard error of means for compounds in each sample. The benefit of having hundreds of compounds per sample is that the certainty of the weighted average is very high, thus a low standard error of the mean. However, due to the overall relatively low replicate number per treatment for each season, further statistical testing would be difficult since the power of these tests is low with low sample size. Despite this, Figure 5 gives us a visual inspection of the seasonal differences of DOM changes in winter and fall incubations which is the main focus of our study.

Changes to revised Manuscript:
L440: Table 2 includes new SD and SEM based on replicates
L476: Figure 5 added error bars
L318: Added forumals for all SD and SEM calculations

3. Lines 32-35: The sentence seems to imply that DOC cannot contribute to carbon sequestration, which is misleading. The authors should acknowledge that DOC can also contribute to carbon sequestration.
We rephrased this sentence to to ensure clarity on the important role that DOC plays in carbon sequestration.

L48: Marine DOM is one of the largest stocks of organic carbon on Earth, contributing to long term carbon storage in the ocean. Particulate organic matter (POM) also aids in carbon sequestration by potentially sinking to the seafloor and transporting carbon from the ocean surface.

4. Line 44: The sources listed for DOM are incomplete. For example, zooplankton grazing can also release DOM from phytoplankton. I recommend referencing "Carlson, C. A., & Hansell, D. A. (2015). DOM sources, sinks, reactivity, and budgets. Biogeochemistry of marine dissolved organic matter, 65-126."
We included the other DOM sources and have revised this sentence accordingly.

L51:  DOM is primarily generated and secreted by phytoplankton during their growth as they release 2 – 50% of the photosynthetically fixed carbon as DOM (Thornton, 2014; Paulsen et al., 2018). DOM can also be produced by zooplankton during grazing and excretion, as well as by bacterial and viral processes such as lysis and excretory release and through the dissolution of particles (Carlson & Hansell, 2015; Riley, 1963; Wagner et al., 2020).

5. Lines 50-55: The authors mention labile and semi-labile DOM but omit recalcitrant DOM, which has a turnover time of millennia. To be thorough and consistent with previous categorization of DOM based on lability, recalcitrant DOM should be included.
We have now modified the text to incorporate recalcitrant DOM and its long residence time.

L59: Labile DOM, which constitutes less than 1% of the overall DOM reservoir, displays relatively short turnover times, typically ranging from hours to days (Hansell, 2013). Conversely, semi-labile and recalcitrant DOM persists in the ocean over more extended time scales, ranging from months to millennia (Fleurs et al., 2012; Hertkorn et al., 2006).

6. Line 145: I do not fully understand why using a headlight with red light can reduce the possibility of biological production. Chlorophyll a absorbs red light.

Chl a absorbs mainly in the blue part of the light spectrum, but indeed also absorbs red light. The reason we do not use green light is that it does not provide as good visibility for the human eye at low intensities. The light exposure (in terms of intensity and time) were minimal in the beginning and the end of the incubation, and therefore we are confident that this light exposure virtually did not impact the incubations. This can be confirmed by almost no increase of phytoplankton cells during winter (change of $-101.5 \pm 23.4$ cells ml-1).

7. Lines 253-254: Why pre-soaking SPE column sorbent with Methanol is needed ?
A pre-soak time can potentially improve the activation of the sorbent and ensure that the solvent fully permeates into the sorbent.

8. Line 259: The DOC recovery is highly variable. Could you explain why?

The greatest variability in DOC recovery was attributed to seasonal differences in the recoverable DOC. This is illustrated in the boxplot Figure S9 added to the supplementary. The highest SPE-DOC recovery, ranging from 74% to 85%, occurred in December t0. This indicates a higher proportion of hydrophobic DOM which SPE-DOC strongly retains (Hertkorn et al., 2013). SPE-DOC recovery decreases during the December incubations, likely due to a reduction in these compounds which is also observed in the overall increase in lability in DOM during incubations. By late winter (February), SPE-DOC recovery declines further (45% , mean SD), reflecting a shift towards more labile material as observed in winter incubations.  Lower recoveries were also observed in September and October, months marked by high biological productivity. During this period, there is a higher proportion of hydrophilic material such as dissolved proteins and carbohydrates from biological activity (Kirchman et. al 2001; Goldberg et. al 2009)—these compounds are not well retained by our SPE sorbent (Grasset et al 2023). Finally, the decrease in SPE-DOC recovery during September and October incubations (40%) also indicate components with low affinity for the SPE sorbent.

Related changes to the manuscript:
L268: updated values to reflect F treatment only since this was the main focus.

Figure S9: Box plot on SPE-DOC % recovery was added to supplementary.

L464: Additionally, the highest SPE-DOC recovery, ranging from 74% to 85%, occurred in December t0 (Fig. S9).L623: Notably, the low SPE-DOC % recovery observed during September and October incubations (Fig. S9) could indicate an increase in hydrophilic fractions of DOM. This points to an increase in proteins and carbohydrates which are present in biologically productive periods (Kirchman et al., 2001; Goldberg et al., 2009). Although these hydrophilic fractions were not extracted in our DOM method, their presence could also

contribute to the increase in POC concentrations observed during the biologically productive period.

L682: However, we also measured an increase in experimental EPS concentrations in the F water during February. This is supported by the low SPE-DOC % recovery observed in February (Fig.S9).

L703: These observations indicate an increase in average lability of DOM and could be partially due to the dissolution of organic particles observed during this period as it can lead to the production of more labile DOM. This is supported by the reduction of SPE-DOC % recovery observed during December incubations (Fig S9) which indicates hydrophilic material at the end of incubation.

9. Line 280: Did the authors try other criteria for formula assignment, such as allowing nitrogen between 0 and 4 ?
The authors would like to emphasize that allowing too many nitrogens leads to false assignments to these formulas. Formulas with a higher number of nitrogens are not very commonly assigned (even in 21T FT-ICR data), and when they are, they are usually very minor peaks. These peaks wouldn't make a large difference to the assigned peak intensities or mass values (in our data, in which they would coalesce with another peak), and even if they were resolved and assigned, would make very tiny differences to the overall metrics.

10. Line 281: Why remove formulas which contain both nitrogen and sulfur, 13C and nitrogen or sulfur and 13C? Could the authors justify it?
Same reason as point 9, to avoid false assignments, these peaks are uncommon and tiny.

11. Lines 395-400 and Figure 5: The changes in DOM parameters like H/C, AImod, MW, and O/C are much smaller than the standard deviation. Although the authors state these small differences are statistically significant, statistical test results should be provided to justify this claim.
We would like to refer the reader to our response in point 2 (in this review).

12. Line 108: should be "characteristics" instead of "character"? Done
13. Line 131: (TPM and PIM). Done
14. Line 141: should be "every two months"? Done

**Reviewer 2:**

1. This is an interesting study and relevant research. However, the manuscript is lacking in statistical testing to support the presented results. Also the presented hypothesis should be clearly addressed at the end of the manuscript.

We would like to thank Reviewer #2 for their interest in our study and thorough comments. The response to point 1 and 2 in Reviewer #1 comments applies here as well.

Following on the comment about statistical evidence, we performed a t-test to test our initial hypothesis; "biologically active periods with higher POM concentrations have a higher potential for aggregation of DOM via adsorption in comparison to the winter period". We performed a t-test on the difference (t1-t0) in POC concentrations in winter vs productive

period (the two groups "winter" and "productive" period had been delineated by the SIMPROF test as shown in the manuscript). For the F treatments, this resulted in a significant ($p = 0.04$) difference in change of concentration between winter and the productive period. This leads us to accept our hypothesis, given we were interested in the seasonal aggregation potential of DOM, represented by the F treatments. For the UF treatments, however, this test was not significant ($p = 0.34$) which is likely due to aggregation, dissolution and biological processes taking place at the same time, which complicate the UF treatment results and its interpretation.

Accordingly, we added the following sentences to the manuscript:

L305: To test our main hypothesis on whether biologically active periods with higher POM concentrations have a higher potential for aggregation of DOM in comparison to the winter period, we performed a t-test on the difference (t1-t0) in experimental POC concentrations between the two biogeochemical periods delineated by the SIMPROF test ("winter" and "productive" period).

L318: Added formulas for standard deviations for between compounds within a sample and standard error of means and standard error of difference between means.

L397: For the F treatment, a t-test revealed significant differences in the change of POC concentrations between winter and the productive period in the F treatment ($p = 0.04$); whereas this difference was not significant for the UF treatment ($p > 0.05$).

L735: Other studies also report synergistic relationships between POC and DOC during summer incubations and antagonistic relationships during freshet incubations (Keskitalo et al., 2022). This indicates a seasonal component to POC-DOC dynamics. Additionally, other incubation studies report large variability in changes in POC and DOC concentrations (Shakil et al., 2022). These findings highlight the challenges in organic carbon measurements as reported elsewhere (Chow et al., 2022) and are likely due to the complex dynamics between POC and DOC such as sticky nature of EPS material which can be adsorbed onto experimental containers (Chen et al., 2021).

L770: Our results confirm our initial hypothesis; that the aggregation potential of POM is higher in the productive period (an average increase of POC of $+1.6$ µM $\pm 0.5$ within 36hr) compared to the winter period where there was an average POC decrease of $-2.55$ µM $\pm 0.8$. Our findings also reveal that the molecular characteristics of DOM during short time frame incubations are influenced by contrasting seasonal conditions at high latitudes, with distinct transformations occurring during the biologically productive and the winter period. It is important to note, however, that the POC and DOC pool did not behave antagonistically during our seasonal incubations, similar to findings in other studies. This underlines the complexity of OM transitions with colloids, gels and biological communities likely playing an important role in these transformations, and more studies are needed to disclose the underlying mechanisms.

L803: added to conclusion: These findings underline that changes during short time frames in the POM and DOM pools are highly subject to seasonal transformations and that biotic and abiotic processes drive these changes. A substantial fraction of POM can potentially originate from the DOM pool through aggregation, although the POC and DOC concentration dynamics indicate more complex processes taking place within the system. Additionally, aggregation within short time frames in the productive period (within 36 h as we have shown) demonstrate that standard POC measurements in the field under periods of high production likely

underestimate actual in situ POC concentrations, as they do not account for the dynamic exchange with the DOM and colloidal pool.

2. I would also encourage the authors to add all the data to the supplement or to a data repository to benefit future research.

We fully support making the data open access and have included DOIs for all datasets, which are now publicly available in a data repository. The code will also be made available with the publication along with mzXML files for running DOM data analysis.

L811: All data are available on the NIRD research data archive (https://archive.sigma2.no). Field data are found under the following DOIs: 10.11582/2024.00096 (CDOM), 10.11582/2024.00116 (POC/PN), 10.11582/2024.00117 (Chl-a), 10.11582/2024.00118 (TPM), 10.11582/2024.00119 (EPS), 10.11582/2024.00120 (nutrients), 10.11582/2024.00131 (protists) and 10.11582/2024.00132 (FCM). Experimental data can be found under 10.11582/2024.00121(POC, FCM and EPS) and 10.11582/2024.00098. (DOC and TDN).

3. Lines 101-102: There are other DOM-POM studies albeit from different environments, see e.g., Attermeyer et al. 2018 (Swedish rivers), Shakil et al 2020 (permafrost thaw streams in Canada), Keskitalo et al. 2022 (Kolyma River).

We would like to thank the reviewer for bringing these studies to our attention and have incorporated them in our revised manuscript.

L106: Studies of the DOM–POM continuum in aquatic environments have mostly been interpreted either from an ecological or a chemical point of view. We are aware of only a few some studies with laboratory and field observations in rivers (Attermeyer et al., 2018; Keskitalo et al., 2022), permafrost thaw (Shakil et al., 2021), Atlantic coast (Riley, 1963) and Pacific coast (Sheldon et al., 1967), or focused on the associated changes in the DOM–POM continuum in boreal peatlands (Einarsdóttir et al., 2020) and rivers (Xu & Guo, 2018).

4. Lines 321-328: Scientific names of species should be in italics here and elsewhere.

This has been fixed.

L357: Most of the year, the microphytoplankton community was dominated by the diatom genus *Chaetoceros* (Fig. S2x). In April, communities were dominated by both *Chaetoceros* sp. and the prymnesiophyte *Phaeocystis pouchetii* (up to 4 x $10^5$ cells mL$^{-1}$), in June by *Chaetoceros filiformis*, the genera *Thalassiosira* and *Pseudo-nitzschia* and in July mainly by *Pseudo-nitzschia*. Unidentified flagellates and *Chaetoceros laciniosus* dominated in August, while September was dominated by cilliates (especially *Strombidium conicum*) and various dinoflagellates that can be mixotroph.

5. Line 356-357: Could you clarify how there can be a decrease in POC in filtered waters where POC has been removed by filtering?

Even after continuous refiltrations, POC can be measured in filtered water (Figure S1 and Valdes-Villaverde et al., 2020). This is likely due to the immediate self-assembly of EPS molecules in filtered water. Therefore, POC cannot be fully "removed" from filtered water and we explain the lower concentrations at t1 with a decrease of remnant POC in the sample. For clarification, we revised this sentence on the manuscript and other instances where we mention dissolution as a likely process:

L391: In winter (December and February), after 36 h of incubation, we measured lower POC concentrations in F water at t1 compared to t0 (a decrease of the mean POC concentration by −2.55 µM ± 0.8 (around -50% relative to t0).

L525: The experiments showed a decrease of POC concentrations in both filtered (F) and unfiltered (UF) water during winter (December and February), suggesting a net dissolution or degradation of particles.

L674: This is also supported by the increase of bacterial activity in the experiment at t1 relative to the start of incubations (t0) throughout the year, and by the fact that a decrease of POC ("particle dissolution") was observed in F and UF water.

The term dissolution was removed from all sub-headers (see response to first comment by reviewer 1).

6. Lines 396-401: So was the statistical significance tested as the authors mention that the results are significant (after explaining about interpretation of signal intensity weighted averages)? It should be made easier for the reader to judge the significance of these results as the changes look so minor. I've read the explanation, but statistical testing could help to understand how meaningful the results are.

We would like to refer the reviewer to answer in point 2 on reviewer #1 responses. Briefly, in the revised manuscript, we only use the word "significant" when we refer to statistical tests and refer to the p value and statistical test applied.

7. Line 441: Typo -> the end of the incubation should be T1 instead of T0.

This typo is fixed.

8. Line 522: Some of the increases in POM could be due to DOM adsorption to particles, or could you clarify if adsorption is included here under aggregation processes?

Yes, DOM adsorption to POM could be occurring. For clarification, we added "adsorption of DOM to particles" to L586.

9. Lines 565-565: Any thoughts why your results were different from Maie et al 2008?

L630: Maie et al. demonstrated that tannin (oxygen rich compounds) aggregation increases with salinity, progressing from freshwater to seawater levels. In contrast, our experiments from September (when aggregation was observed) do not show a decrease of these highly oxygenated DOM compounds, likely due to a limited tannin source in this region as shown by the low seasonality in the tannin region of the van Krevelen diagrams (Figure S8). Instead, other studies (Riedel et al., 2016) show the preferential biological degradation of low oxygen compounds which could also be occuring in our incubations as indicated by high bacterial actrivity and removal of low O/C formulas.

10. Line 573: I assume the abbreviation LMW refers to light molecular weight, but it should be defined here.

LMW refers to low molecular weight compounds and is now added in the text (L643).

11. Line 617: So the difference between experimental and in situ POC concentrations is the sieving step? Could these differences be more clearly mentioned and also presented in a supplementary table for example (I see that the information exists in separate figures in the supplement, but it is rather difficult to compare).

This sentence was added:

L693: Note that the water collected for the UF treatment was sieved through a 90µm mesh, while the water collected for field measurements was not sieved.

Table S1 was added to the supplementary, and referred to in the manuscript:

L691 "Experimental POC concentrations at the start of incubation (4.7 – 5.3 µM in F, and 5.6 – 6.0 in UF water; Fig. S4) were similar to field POC concentrations in winter (around 4 µM, Fig. S2y), which suggests that particles were of extremely low abundance and size during the winter period (Table S1)."

12. Lines 633-636: The t-peaks should be mentioned already in the methods. Could you also clarify to what the statistics refer to on line 635 (e.g. which test and which samples exactly). Were the t-peaks not observed during other seasons than winter?

L293: T-peaks have been added to the methods section of the manuscript.

L711: T-peak changed significantly in December but did not change significantly during September, October and February incubations based on independent t-test results on each month's replicates for time 0 versus time 1 (four t-tests at the 5% significance level).

L79 in supplementary: Figure S7 including all t-peaks in all months was added to supplementary.

13. Lines 682-716 (conclusion and outlook): I suggest focusing on results and conclusion from your study while leaving out comparisons with other studies. Furthermore, this section is lacking a clear answer to your original hypothesis and the subsequent questions that were laid out in the introduction (on lines 104-107).

Following this suggestion, the conclusion was rewritten in a way that reflects the findings of the study in a clearer way. In addition to removing the following sentence from the conclusion: "This contrasts with other observations that show aggregation during winter in a temperate region (Riley, 1963) and is possibly because temperate systems are not light-limited and primary production (and with that, EPS exudation) can take place throughout the year, even if it is reduced. References were removed, as these were referred to earlier in the discussion and introduction to avoid repetition. See updated conclusion (L765).

Figures:

Figure 2. Nice and useful figure summarising the sampling and incubation set-up.

Thank you for this feedback.

Figure 4. Add statistical testing on the changes in POC and DOC and different parameters before and after incubations. Were the changes significant? Also adding a panel of total organic carbon (DOC+POC) would be interesting.

A panel for total DOC+POC was not added, because it follows the same patterns as DOC as the changes in POC are minimal compared to DOC. The changes in POC or the variation will not be reflected in this panel.

We performed t-test on DOC and POC concentrations at t0 vs. t1 for each month. For DOC, the differences between t0 and t1 were significant in September ($p < 0.05$) and December ($p < 0.05$) incubations. However, for October ($p = 0.07$) and February ($p = 0.7$), the differences were not statistically significant. Regarding POC, the changes between t0 and t1 were also not significant ($p > 0.05$). The standard deviations in Figure 4a) demonstrate that the range of change in POC concentrations is wide, except during the winter months. DOC and POC quantification can have challenges due to its complex nature (Chow et al., 2022; Gardner at al., 2003) and we recommend increasing the sampling size in future studies. However, this was not feasible within the seasonal scale of our experiments and with the range of different parameters taken. Therefore, in our revised manuscript, we focus on the contrast between winter and the productive period of DOM characteristics and POC. To highlight this, as mentioned in point #1, we conducted a t-test on the difference (t1-t0) in POC concentrations in winter vs productive period. This resulted in a significant ($p= 0.04$) difference in change of concentration between winter and the productive period.

Moreover, increasing the incubation time could possibly have resulted in more distinct differences between t0 and t1; however, we wanted to explore the immediate changes in the OM pool, and many studies have shown that aggregation occurs within shortest time scales (hours). Longer timescales of water body incubations would also be less reflective of ambient conditions and might lead to bottle effects.

Related changes in the manuscript:

L305: To test our main hypothesis on whether biologically active periods with higher POM concentrations have a higher potential for aggregation of DOM in comparison to the winter period, we performed a t-test on the difference (t1-t0) in experimental POC concentrations between the two biogeochemical periods delineated by the SIMPROF test ("winter" and "productive" period).

L3976: For the F treatment, a t-test revealed significant differences in the change of POC concentrations between winter and the productive period ($p = 0.04$); whereas this difference was not significant for the UF treatment ($p > 0.05$).

L552: Although the change in POC concentration was small (between -3.10 – +2.28 µM) and varied across individual tanks (Fig. S4; most likely due to the short incubation time), we show for the first time the contrasting seasonality of this aggregation process.

Figure 5. Why has standard deviation/error not been included in this figure?

We have now added error bars for the standard error of the difference of means to Figure 5 and apologize that this was not added with the initial submission.

Figure 6. This figure is difficult to read. The font sizes should be larger, especially the axis titles. The green color is explained in the caption, but I'd suggest adding that to the legend as well. Why are some bars separated from others with black lines but some aren't? Why do some bars have black horizontal line at the top of the bar while others don't? Some of the panels have lines next to the axis while others don't, I suggest re-making this figure or using a different figure type.

We appreciate the reviewers feedback on this figure and have remade Figure 6 to a side-by-side histogram with two distinct colors for clearer comparison.

Figure S5: We also updated the histogram plots for UF treatment in the supplementary and put the start and end of incubation side-by-side to improve visibility.

Figure 7a. Could you add to the legend the meaning of the different colors (green, blue).

A legend has been added.

Table 3. Wouldn't it be more meaningful to compare the changes (in %) in POC during incubations to the experimental POC t0 concentrations and then after calculate how much that % is from *in situ* concentrations?

We would like to ask for clarification on this comment.

---

## Author Response (AR2)

Few final comments (line numbers refer to the track changes document):

The authors have made good improvements to the manuscript according to the review. As I point out in my comments below it would be useful to add extra clarity on the standard deviations used for the DOM metrics data (as the other reviewer and I pointed out already in our initial comments). I see the reply of the authors regarding this, however, to me it is not fully clear how they have reduced the standard deviations so dramatically.

1. Line 191: Presumably these filters were also 0.7 µm?
We added the pore size to the text: 'GF/F (Whatman®, 0.7 µm, pre combusted)'.

2. Line 436: These standard deviations (in Table 2) do not seem 'high' as described in this sentence.
This section refers to Table S2 in the supplementary material (not Table 2 in the main text). Please see response for next section for changes made to address this.

3. Table 2. The authors seem to have been able to decrease the standard deviation by an order of magnitude or fully remove any deviation for the DOM metrics by using replicate samples instead of the compounds. I am not sure I fully understand how these differ from each other here. Any extra clarification in the text could be useful for the readers.
We agree that including explanations for both standard deviations in the main text could be confusing and to add to the confusion, the two standard deviation equations use the same symbol, N, to describe two different things. To address this, we have changed replaced the two N symbols to different symbols (F and N). F represents the number of formulas within a sample and N the number of samples within a treatment. Additionally, the text describing Table S2 was moved to the supplementary material in the caption text for Table S2. We also reference the equations for the respective standard deviation in the caption of the table.
The weighted standard deviation, SDw (Equation 3, Table S2) in the supplementary material describes the variability of formulas within a sample, and the standard deviation, SD, (Equation 2, Table 2) in the main text explains the variability between treatment means. The high variability of formulas within a sample (SDw) help explain our rationale for interpreting small changes between means as large changes among hundredths of formulas per sample. We hope that these changes will provide better guidance for the reader.

New text on the supplementary Table S2:
L106: 'The weighted standard deviation (SD$_w$, Equation 3) and the standard error of the weighted mean (SEM, Equation 4) are computed for each sample using the DOM metrics associated with the molecular formulas within that sample. The variability in metric values among the many identified formulas is reflected in the standard deviations. The benefit of the large number of formulas for each treatment is the high certainty in the mean which is also shown by the low standard error of mean.'

New text for Table 2 in manuscript:
L439: 'The standard deviation (SD) is computed (Equation 2) for the start (t0) and end (t1) of incubation of the 36 hours incubation for the filtered (F) fjord water treatment (N = 3). The standard error of the difference of means (SEM$_{x_1-x_2}$) is computed (Equation 5) for each experiment.'

We also added the line:
L430: 'A table presenting the individual mean values per sample for each DOM metric is provided in the supplementary material (Table S2).'

Additionally, we replace the symbol for N with F in the weighted standard deviation equation in the methods section:
L311: The equation for weighted average for DOM metrics (H/C$_{wa}$, O/C$_{wa}$, AI$_{mod\ wa}$, MW$_{wa}$) is shown below. Here, $I_i$ is the signal normalized intensities for a given formula and $A_i$ represents the DOM metric value for that formula and F is the total number of formulas per sample.

$$wa = \frac{\sum_{i=1}^{F} I_i \cdot A_i}{\sum_{i=1}^{F} I_i}$$

(1)

The standard deviation shown in Table 2 is calculated for the mean DOM metric in each treatment. Here, $wa_i$ is the intensity weighted average for each sample and (N) represents the total number of samples per treatment and $\overline{x}$ is the sampling mean for the treatment.

$$SD = \sqrt{\frac{1}{N-1} \sum_{i=1}^{N} (wa_i - \overline{x})^2}$$

(2)

Additionally, the weighted standard deviation ($SD_w$) for compounds in each sample is estimated by the following equation and shown in Table S2. Normalized intensities for a given formula are represented by $I_i$. DOM metric value is represented by $A_i$, and F is the total number of formulas per sample. The weighted average, $wa$, equation is the same as Equation (1).

$$SD_w = \sqrt{\frac{\sum_{i=1}^{F} I_i (A_i - wa)^2}{\sum_{i=1}^{F} I_i}}$$

(3)

The standard error of a single mean's true estimated value (and not the distribution of the population, which is estimated by SD) is approximated (as the SEM) by the following equation where the weighted standard deviation, $SD_w$, is divided by the square root of the number of identified formulas, $F$.

$$SEM = \frac{SD_w}{\sqrt{F}}$$

(4)

The following equation is used for calculating standard error of the difference of means ($SEM_{x_1 - x_2}$) between treatments as shown in Table 2. Here, the standard deviations are divided by the number of samples, N, for each treatment.

$$SEM_{x_1 - x_2} = \sqrt{\frac{SD_1^2}{N_1} + \frac{SD_2^2}{N_2}}$$

(5)

4. Supplement line 83-84: How is the n only three for the t-test as seemingly there are so many points (n=53-73) shown in the figure? I'd advise also better practise in reporting t test results: include the t value and degrees of freedom in addition to the p value.

Thank you for this point. The individual highlighted points on the van krevelen refer to each individual tpeak formula for that sample. We ran t-test on the total number of tpeaks reported per replicate (n=3) per treatment. A total of four t-test with a 95% confidence level. We have added the t-values and degrees of freedom for each t-test reported and corrected a typo on the months.

L700: 'Additionally, there was a significant decrease of a group of low H/C compounds referred to as 'terrestrial peaks' (t-Peaks) in February (t-test, p = 0.04, t-value: 2.96, df = 4) and October (t-test, p = 0.04, t-value: 3.02, df = 4). T-Peaks are a group of compounds that are commonly present in vastly different rivers as reported by Medeiros et al. (2016; Fig. S7). Removal of these compounds could contribute to the increase in average H/C$_{wa}$ ratios observed in February and October incubations. This suggests a potential degradation of t-Peak compounds during February and October, in contrast to September (t-test, p = 0.08, t-value: -2.31, df = 4) and December (t-test, p = 0.09, t-value: 2.21, df = 4) when t-Peaks did not significantly change during the incubation (Fig. S7).'